# Three-dimensional spike localization and improved motion correction for Neuropixels recordings

**Julien Boussard**$^*$    **Erdem Varol**$^*$    **Hyun Dong Lee**    **Nishchal Dethe**    **Liam Paninski**

Department of Statistics and Neuroscience, Center for Theoretical Neuroscience, Grossman Center for the Statistics of Mind, Zuckerman Institute, Columbia University; *-equal contribution

## Abstract

Neuropixels (NP) probes are dense linear multi-electrode arrays that have rapidly become essential tools for studying the electrophysiology of large neural populations. Unfortunately, a number of challenges remain in analyzing the large datasets output by these probes. Here we introduce several new methods for extracting useful spiking information from NP probes. First, we use a simple point neuron model, together with a neural-network denoiser, to efficiently map single spikes detected on the probe into three-dimensional localizations. Previous methods localized individual spikes in two dimensions only; we show that the new localization approach is significantly more robust and provides an improved feature set for clustering spikes according to neural identity ("spike sorting"). Next, we denoise the resulting three-dimensional point-cloud representation of the data, and show that the resulting 3D images can be accurately registered over time, leading to improved tracking of time-varying neural activity over the probe, and in turn, crisper estimates of neural clusters over time. Open source code is available at `https://github.com/int-brain-lab/spikes_localization_registration.git`.

## 1   Introduction

Neuropixels (NP) probes are dense linear multi-electrode arrays that enable the simultaneous observation of hundreds of neurons across multiple brain areas. Since their introduction in [13] (see also [27]), they have rapidly become an essential neurotechnology, deployed in hundreds of labs around the world. A number of challenges remain in analyzing the large datasets output by these probes. The basic goal is "spike sorting" — i.e., to detect action potentials ("spikes") and assign these spikes to individual neurons. Despite significant effort in the field [4, 12, 21, 27], current spike sorters for NP data have trouble tracking temporally non-stationary data and accurately sorting small spikes [28].

One major advantage of NP probes is their spatial density: the high spatial resolution of electrodes on the probe implies that each extracellular spike will typically be detected at multiple sites, providing an opportunity to "triangulate" the location of each spike. If effective, this spike localization can be helpful for multiple downstream tasks: spike locations serve as useful low-dimensional summarizations of the high-dimensional spatiotemporal spiking signals, which can in turn be visualized and used as features for clustering and tracking as the probe undergoes small motion relative to the brain.

Several spike sorting algorithms localize spikes using a simple center of mass (CoM) method. More precisely, for each waveform, they select a set of channels, and compute the weighted average of the selected channels' positions, where the weights are given by an estimate of the amplitude of the spike on each channel. This fast, cheap method gives informative location estimates for spikes in multi-electrode array (MEA) recordings where long dendritic or axonal signals are not prevalent [23, 20, 10] (cf. the primate retina [17], where long axonal signals make localization approaches less useful). However, by definition, the CoM approach localizes spikes inside the convex hull of the electrodes, which is especially problematic for Neuropixels recordings, due to the long and thin shape of the electrodes. Hurwitz et al. [11] propose an improved localization method that provides location

35th Conference on Neural Information Processing Systems (NeurIPS 2021).

estimates that can lie outside of the probe, but the estimates are still (like the CoM method) restricted to the two-dimensional plane spanned by the electrode locations. In addition, several approaches have been developed for localizing *templates* (averages over many spikes, post spike-sorting) in three dimensions, using biophysical models of varying sophistication [26, 25, 8, 5]. Finally, triangulation approaches based on point-source [6, 15] or dipole models [19] have been proposed for smaller-scale tetrode recordings, in which spikes are only detected on four electrodes, limiting the resolution of the resulting localization in the case of small, noisy spiking units.

In this paper, we build on this previous work, making four key contributions:

1. We introduce a denoise-then-triangulate approach, based on a simple point neuron model, that localizes single spikes in three dimensions.

2. This improved three-dimensional localization in turn enables better spatial clustering of distinct neural units characterized by unique action potential waveform shapes.

3. Our localization further enables better motion estimation in recordings with drift.

4. Our approach is robust to different probe geometries. We demonstrate its utility and performance in both Neuropixels 1.0 and 2.0 probes.

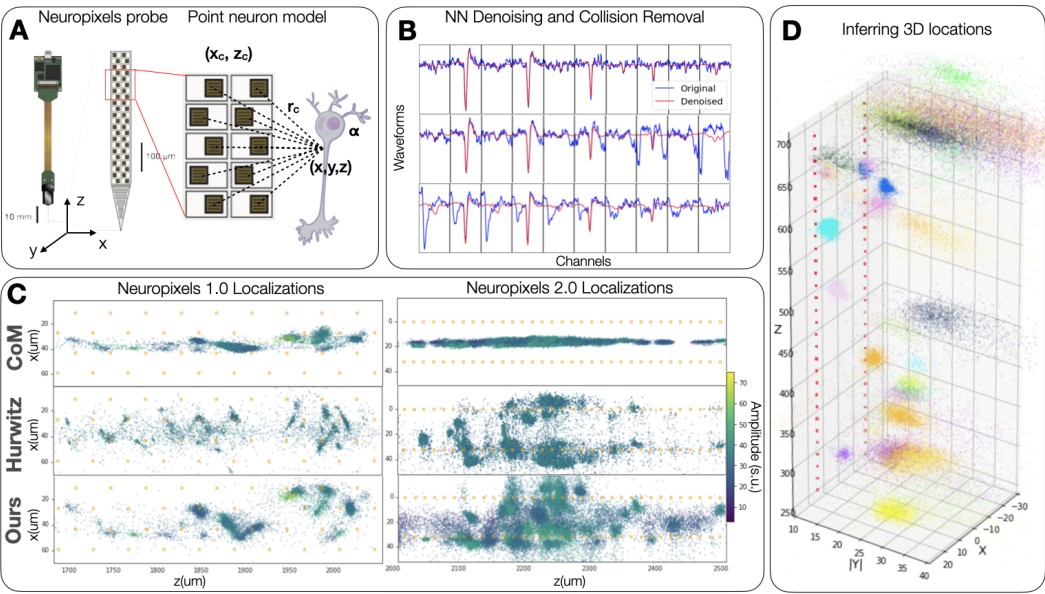

Figure 1: **Overview of the proposed localization technique.** (**A**) The peak-to-peak voltage amplitude attenuation recorded at each channel of the Neuropixels probe is modelled as approximately inversely proportional to the distance to a spiking unit point source. The 3D spatial location of the spiking unit is then inferred by essentially solving a triangulation problem (minimizing a least-squares loss to all the channels that detected the spiking event). (**B**) Since the triangulation depends on the peak-to-peak amplitudes of spiking events, we denoise the amplitudes using a neural net and further disambiguate distinct units that have spiked at the same time (collision removal). Each row shows a single spike event (original and denoised), captured on multiple nearby electrodes. (**C**) We apply our method to spatially map the localizations of over one million spiking events in both Neuropixels 1.0 and Neuropixels 2.0 recordings. Our localization is compared with the state-of-the-art localization method proposed in [11], as well as the baseline method of inferring position using center of mass (CoM) of channel positions weighted by the recorded amplitude. Spikes are colored by maximum amplitude, normalized by the recording baseline noise level ('standardized units' / 's.u.'). Yellow corresponds to high and dark blue to low amplitude. (**D**) Our localization yields groups of spiking events that disperse in 3D space and cluster based on amplitude. Colors correspond to clusters of waveforms found by YASS [17], for visualization purposes only. Orange squares represent recording electrode positions. Datasets from https://github.com/flatironinstitute/neuropixels-data-sep-2020/blob/master/doc/cortexlab1.md [27]; see **video-figure-1.mp4** for a Datoviz [24] visualization.

## 2 Methods

### 2.1 Point neuron model

A neuron's spike is the result of a quick change in its membrane potential due to the shift of its charged particles from one side of the membrane to the other. This movement of charge creates an electric field outside of the neuron [14], and Neuropixels electrodes record the potential of this field, at multiple locations along the probe (**Fig. 1A**). We use a simple point-source model for these voltage differences at distance $r$ from the cell:

$$V \simeq \frac{\alpha}{\sqrt{b^2 + r^2}} \simeq \frac{\alpha}{r},$$

where $\alpha$ represents the cell's overall signal magnitude, and the parameter $b$ is fixed independently of the cell. This point-source model has been previously applied in tetrode recordings [6, 15], and could be replaced with more detailed models (dipole, ball-and-stick, etc) if desired.

### 2.2 Spike denoising and collision suppression

The triangulation approach described below relies only on the amplitude of the spike on each electrode. In practice we measure this amplitude using the peak-to-peak (PTP) value (maximum minus minimum voltage over a short temporal interval). In the presence of noise, the PTP is biased upwards (since noise will increase the maximum and decrease the minimum signal), and this bias depends on the underlying signal strength (the PTP of small spikes is more biased than large spikes). Furthermore, collisions from near-synchronous spike events are prevalent in dense multi-electrode data [17], and even small collisions can lead to inaccurate location estimates if not accounted for properly.

Therefore, it is critical to incorporate a denoising / collision-suppression step prior to triangulation. We use the neural net denoiser implemented in YASS [17] (retrained on NP data), which takes advantage of the waveform's spatiotemporal signature, to suppress both noise and collisions (Note that simple image denoising methods such as [16, 7, 3] do not serve to suppress collisions, and are therefore less effective here.). **Fig. 1B** shows three examples of waveforms (in blue) and their denoised version (in red). The neural net denoiser removes noise and collided spikes, allowing us to successfully estimate each spike's PTP amplitude; see the Appendix for further details.

### 2.3 Spike localization

Given the denoised PTP amplitudes obtained above, we can proceed with localizing the origin of each waveform with respect to the multiple electrodes on the NP probe. We will denote $\{x, y, z\}$ the coordinates of the point neuron, and $\{x_c, z_c\}$ the planar coordinates of each NP electrode, with $y_c$ set to zero by convention. Here we exploit the fact that each spike is detectable on multiple channels simultaneously: i.e., we have multiple observations of the form $\text{ptp}_c \simeq \frac{\alpha}{\sqrt{b^2 + r_c^2}} = \frac{\alpha}{\sqrt{(x-x_c)^2 + (z-z_c)^2 + y^2 + b^2}}$, and our goal is to infer $x, y, z$, and $\alpha$. Thus, for each spike and selected set of channels $\mathcal{C}$ around the "main" channel for each spike (i.e., the channel with the largest PTP amplitude), we want to optimize:

$$\sum_{c \in \mathcal{C}} \left( \text{ptp}_c - \frac{\alpha}{\sqrt{(x - x_c)^2 + (z - z_c)^2 + y^2}} \right)^2.$$

See Appendix for full optimization details. (One important note here: due to the planar NP electrode geometry, the sign of $y$ is not recoverable, i.e., we can't infer whether a spike is in front of or behind the probe, even if we can infer the orthogonal distance from the probe plane.) **Fig. 1 (C)** shows examples of inferred {x,z} locations of detected spikes in both NP1.0 and 2.0 probes (which have different electrode layouts), colored by max PTP. Panel (D) shows an example of 3-d {x, y, z} localization for the NP2.0 data.

### 2.4 Point cloud to image denoising

The estimated localizations are a point cloud in continuous space (**Fig. 2A**). To enable the use of image registration techniques, we convert the point cloud representation to an image representation by generating spatial histograms where pixel location and intensity capture the spatial coordinates of

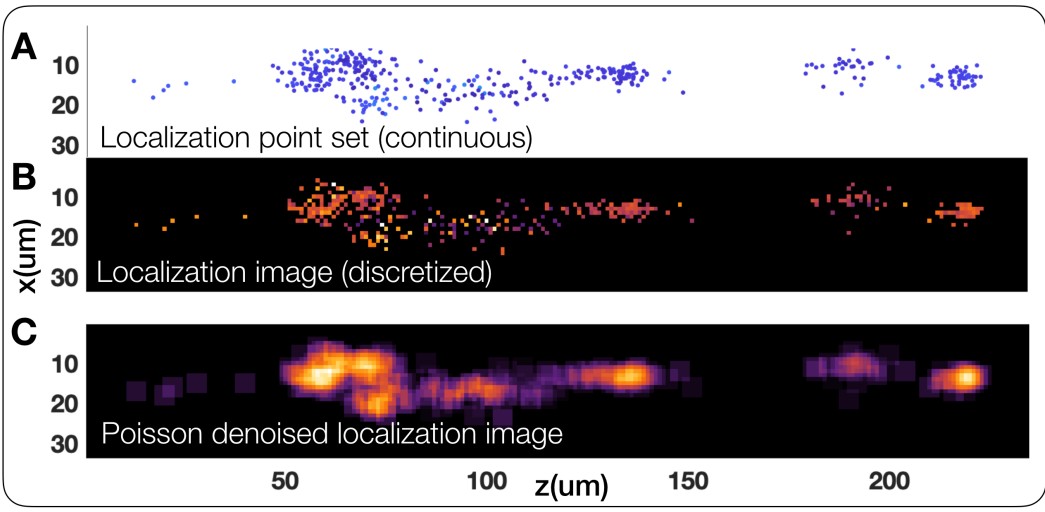

Figure 2: **Poisson denoising point sets of localizations yields interpretable images of spiking unit locations**. (**A**) The estimated spiking unit localizations are a sparse point set in continuous space. (**B**) By binning spike locations in a spatial histogram, we generate a localization "image" from the point set input, where pixel intensities that denote the average amplitude in positions in space roughly follow a Poisson distribution. Note that this image is also sparse. (**C**) Modelling the noise distribution of this image as Poissonian enables the use of Poisson image denoising techniques [18]. The resulting de-sparsified pixelwise representation of spiking unit locations is more amenable to downstream tasks such as image registration and visualization (see **Fig. 4, 5**).

spikes and their mean amplitudes (**Fig. 2B**), following [27]. Note that this image representation is sparse and resembles a speckled Poisson image [18], due to the limited number of spikes that are captured in each spatial position. In order to compute more accurate motion estimates, it is crucial to appropriately denoise this image representation. We adopt a Poisson denoising technique [18], with three steps. First, we apply the Anscombe root transformation [1] to the histogram. Second, we use a Gaussian denoiser (e.g. Block-matching and 3D filtering [7], or non-local means [3]) on the transformed histogram image. Third, the denoised signal is obtained by applying an inverse transformation to the denoised transformed histogram. **Fig. 2C** demonstrates the output for one second of time-binned data: Poisson denoising yields a more continuous pixelwise representation of the spatial histograms, which satisfies the pixel-wise continuity assumptions of image registration algorithms [9] and enables a more robust estimate of displacement.

## 2.5 Motion inference and registration

Given a set of dense Poisson denoised localization images for each second of recording, we estimate relative motion between frames by treating the set of images as a time-series video and use image registration techniques. Several techniques have been recently introduced to infer motion in Neuropixels recordings. The method proposed in [27] follows a non-rigid template based approach, where an average image of localizations is used as an anchor to register individual localization images using a phase-correlation based displacement estimator [9]. In contrast, the method in [29] follows a decentralized non-rigid approach, treating each localization image as its own template performing decentralized pair-wise motion estimates. This method has been shown to be more robust than the template based approach and we utilize it here to infer displacement both within the NP plane ($\mathbf{z}$, $\mathbf{x}$) and orthogonal to this plane ($\mathbf{y}$) for the localization images. For other methods that yield only planar localizations, we just estimate vertical ($\mathbf{z}$) and horizontal displacement ($\mathbf{y}$). The motion estimates for un-denoised and Poisson denoised localization image sets are shown in **Fig. 5D**. Once we estimate motion estimates for each time frame, we then correct for motion by volumetrically (or planarly) translating each image by the negative of the local non-rigid displacement that is estimated for that frame at that location on the probe, bringing all frames into alignment. Example average images before and after alignment can be seen in **Fig. 5H**.

## 2.6 Point-cloud registration

In addition to the image-based registration approach described above, we also experimented with direct point-cloud registration approaches, which skip the formation of an intermediate image representation and are therefore amenable to applications involving higher-dimensional spike featurizations. Representing the spikes in each second of data as a point set, we remove outliers [17] and compress the point cloud via simple hierarchical clustering, parameterized by a single maximum merge distance variable. Then, for each pair of compressed point clouds, we run the iterative closest point (ICP) algorithm [2] to find the displacement in the $z$-direction that minimizes the distance between the clouds, weighted by amplitude and number of spikes. To avoid poor local optima, we utilize a coarse grid search in $z$ to initialize the algorithm; see the supplement for full details. The two motion inference methods lead to comparable displacement estimates on the datasets presented in this paper.

# 3 Results

## 3.1 Comparing localizations and clustering

**Fig. 3A**'s first three panels show the $\{\mathbf{x}, \mathbf{z}\}$ inferred location of detected spikes in 200 seconds of a Neuropixels 2.0 recording, using center of mass (CoM), the Hurwitz et al. [11] method and the new proposed method. (For the Hurwitz et al. method, channels within 35 $\mu m$ from the main channel are included and amplitude jitter is set to 0 $\mu V$.) While CoM does not allow us to separate waveforms along the $\mathbf{x}$-axis or localize outside the convex hull of the electrode, Hurwitz et al. and our method induce spike clusters that appear very different. Hurwitz et al.'s clusters tend to be more isotropic and often appear "paired" on both sides of the probe, while our method leads to differently sized clusters, without pairing. Low amplitude clusters are more spread out than higher amplitude clusters, which is expected as noisier amplitude measurements will lead to noisier localizations.

Our method infers two additional features, $\mathbf{y}$ and $\alpha$. **Fig. 3A**'s two rightmost columns show scatter plots of $\mathbf{z}$ vs $\mathbf{y}$ and $\alpha$. From the clusters in boxes 2,3,4 in the right column, it appears that $\alpha$ can separate waveform clusters more strongly than $\mathbf{y}$, suggesting that this variable contains useful information about the spatial shape of each spike; see the Appendix for further exploration.

The four boxes in blue, green, pink, and red highlight the different spike locations corresponding to similar z-location for each method. For the spikes inside these boxes, we clustered Hurwitz et al. $\{\mathbf{x}, \mathbf{z}\}$ locations, and our $\{\mathbf{x}, \mathbf{z}, \alpha\}$ features using Gaussian Mixture Model with 2 or 3 components. The corresponding waveforms are shown in **Fig. 3B**. The new method provides visually improved waveform clustering here; the separation induced by $\alpha$ is also coherent with the waveform shapes.

A striking difference between the **Fig. 3B** left and right columns' waveforms is that the right column waveforms appear much less noisy. Most of Hurwitz et al. waveforms appear not centered, but really are collided waveforms representing distinct neural units that are poorly disambiguated. In panels 1, 2, and 3, the Hurwitz et al. method localizes small amplitude spikes collided with large amplitude ones in the same region as the largest amplitude spike. This highlights the importance of using the individual-waveform neural network denoiser. In the supplementary material we contrast the results of all three localization methods using denoised vs raw waveforms. Finally, we find that the Hurwitz method sometimes experiences instabilities in the localization leading to separated clusters in the estimated localization space that do not correspond to separations in the raw data. **Figure 3** panel 4 illustrates this idea, as many spikes of similar amplitude are located by Hurwitz et al. method on either the right or left side of the probe (yellow or red clusters).

## 3.2 Visualizing spike densities after motion correction and registration

After localization and image-based denoising and alignment (as described in the Methods), we compute the average post-aligned image to visualize the volumetric spatial layout of distinct spiking units in the vicinity of the probe. The particular recording that we analyze has been inserted through the cortex, hippocampus, and thalamus of a mouse brain (see **Fig. 4A** for an approximate insertion positioning). The average localization image before and after motion correction and registration for the full probe length is shown in **Fig. 4B**, where we can make out three spatially distinct populations of neurons. Using the depth information and the approximate insertion order, we can then identify and annotate the specific anatomical regions that the neurons reside in and zoom in to observe

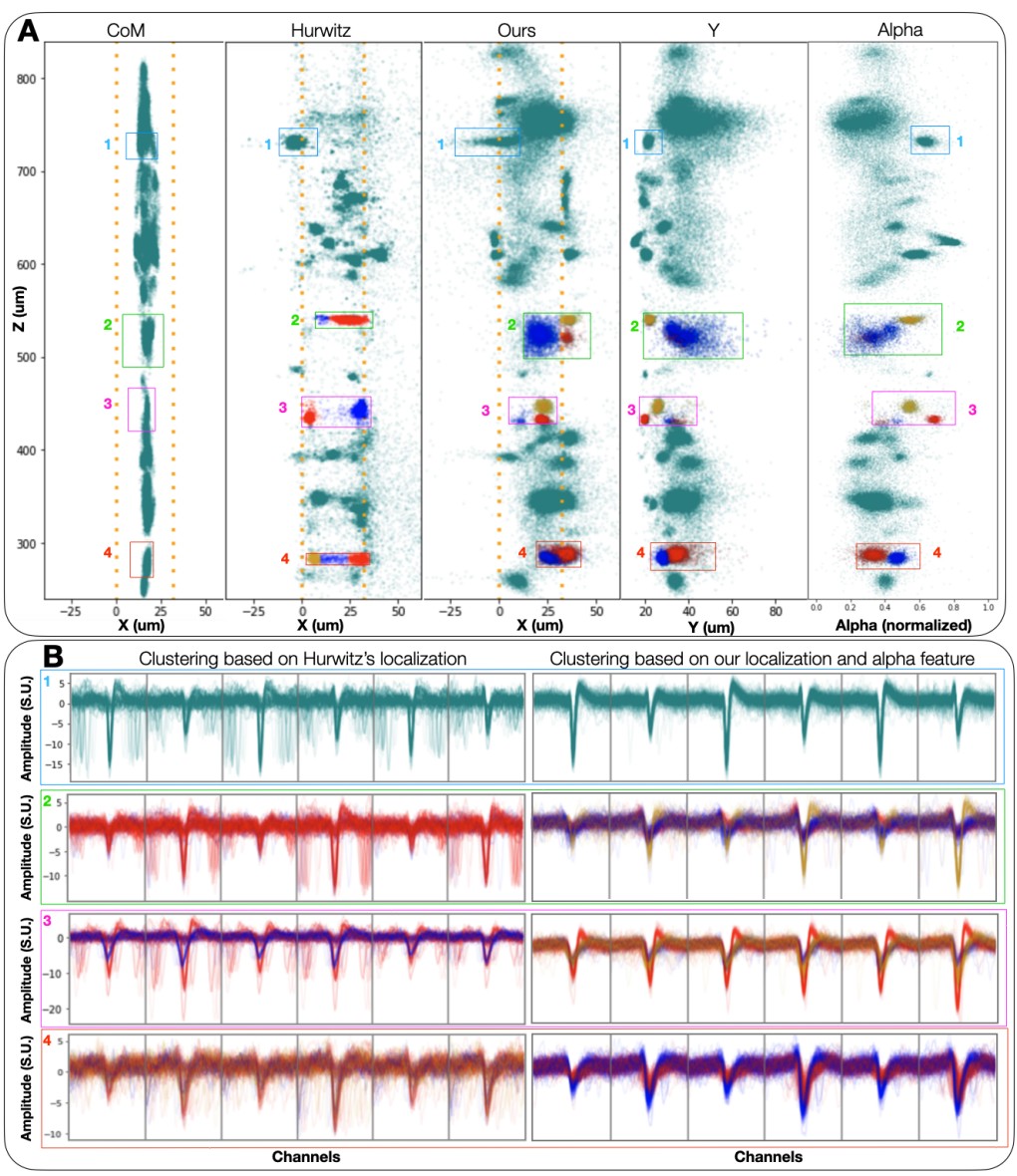

Figure 3: **Inferred 3D spatial features yield improvements in waveform clustering.** (**A**) {x, z} locations of spikes detected from 200 seconds of NP 2.0 recording, inferred by the center of mass baseline (left), Hurwitz et al.'s [11] state-of-the-art method (second to left), and our method (middle). The two scatter plots on the right of the figure represent z vs. y and $\alpha$, where y and $\alpha$ are additional features our method learns. The four colored boxes frame spikes that are shown in panel B, and the yellow, blue and red color of the spikes represent GMM cluster assignments, using {x, z} and {x, z, $\alpha$} as features for Hurwitz method and our method, respectively. The number of clusters has been chosen by hand to reflect the shape of the point clouds in each box. Orange squares represent the position of NP 2.0 recording channels. The boxes in each column are not the same size, as we are trying to match spikes localized in the same z-area, which depends on the localization method. (**B**) Waveforms corresponding to each box and cluster, for both Hurwitz et al.'s (left) and our method (right). (We don't attempt to cluster based on CoM features, due to the lower resolution of the CoM output.) Note that there is no clear one-to-one correspondence between the left and right waveforms, as our denoising / collision removal can change localization drastically. The many non-centered waveforms in the left column show collided spikes that have not been localized properly as Hurwitz et al. method lacks collision removal. In addition, we find that the Hurwitz et al. approach often spatially separates similar waveforms, while failing to isolate different units. On the other hand, our location-based clusters correspond to visibly apparently separate units, without corruption from poorly localized collided spikes. Moreover, the additional feature $\alpha$ can correctly separate different waveforms when {x, z} features alone cannot (red box). See supplementary material for further details.

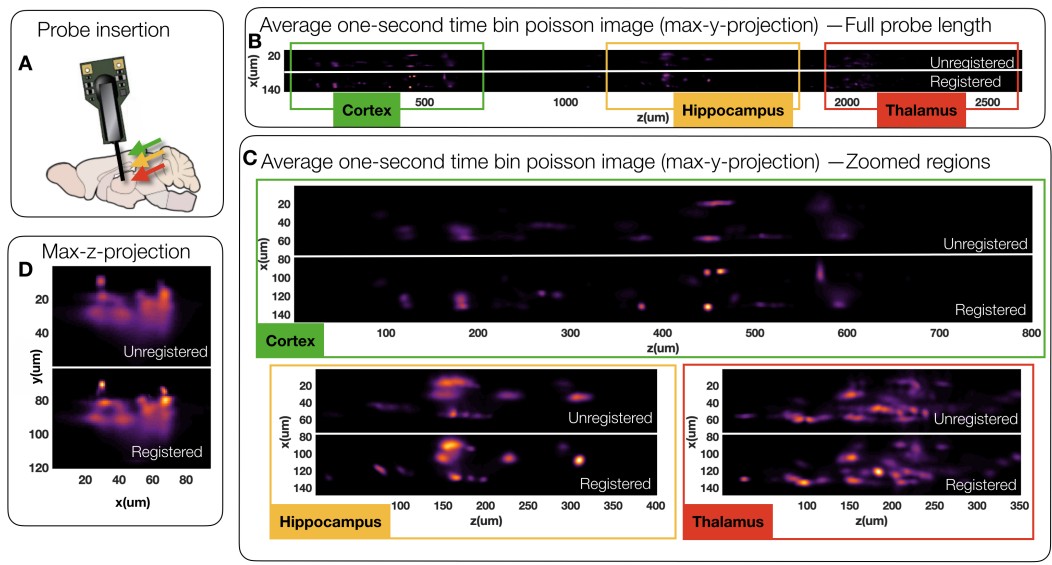

Figure 4: **Averaging motion corrected and registered localization images enables a crisp visualization of spiking density.** (**A**) We analyze an NP 2.0 recording that has been inserted in the regions of cortex, hippocampus, and thalamus of a mouse brain. The average of thousand one-second localization images after motion correction and registration yields a crisp mapping of neural units in three distinct anatomical regions. Max y-dimension visualization is shown in panels (**B,C**) and max-z-dimension visualization is shown in panel (**D**). (**C**) Zooming into cortical, hippocampal, and thalamic depth regions shows a significant improvement of the crispness of the average localization image after registration. Note that motion effects appear as long blurry streaks in averaged images, whereas well corrected motion yields globular and bright individual clusters. (**D**) Maximum projection along the z-dimension emphasizes localization quality as a function of depth and shows a crisper localization of units near the probe and more uncertain localizations farther away (in depth), owing to higher localization uncertainty for more distant, smaller spikes; this effect is also visible in **Fig. 3A**. See supplementary material and video (**video-figure-4.mp4**) for a detailed comparison of unregistered/registered frames.

their densities **Fig. 4B,C**. Note that there is a higher density of spiking units in the thalamus (**Fig. 4C-bottom-right**). By contrasting the unregistered average images with the registered ones, we can see the effects of motion, such as blurred streaks of cell shapes, are mitigated after registration, yielding sharper images of neural populations. Also, visualizing the average image projected along the length of the probe (**z-axis**) shows that while registration does improve sharpness of localization in units close to the probe, "deeper" units that are further away from the probe (and therefore have lower amplitude) remain localized with more uncertainty, as visible also in **Fig. 3A**.

### 3.3 Motion inference and registration evaluation

To further quantify whether our localization and Poisson denoising enable better motion estimation and image registration, we compare the resulting motion estimates and registration metrics using the localization images generated with our method versus the localizations from Hurwitz et al. [11] and the center-of-mass localizer (CoM). To evaluate registration quality, we utilize a Neuropixels 2.0 recording with physically introduced exaggerated motion using an actuator motor (**Fig. 5A,B**). The details of this publicly available recording are shared in [27]. The motion estimates based on all three localization methods and using raw versus denoised localization images can be seen in **Fig. 5D**. The main conclusion is that Poisson denoising significantly improves the motion estimation jitter for all methods compared to utilizing the raw localization images. Furthermore, our localization method affords the estimation of an additional dimension of motion, namely along the depth dimension (**y-axis**) to yield more refined motion estimates. To evaluate motion estimation quality along the length axis (**z-axis**), we provide raster plots before and after registration using all three localization

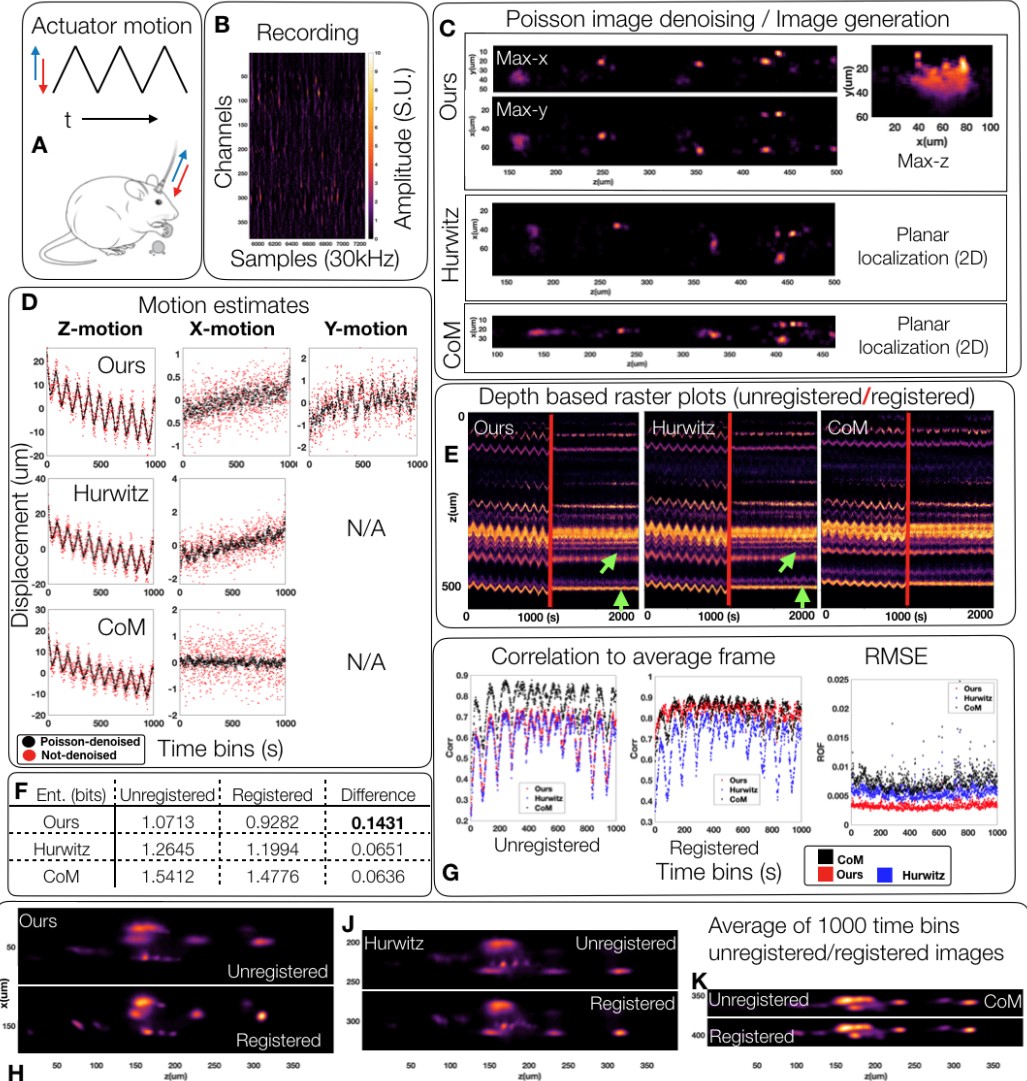

Figure 5: **Improved localization enables better motion correction and registration of Neuropixels recordings.** (**A, B**) We analyze a mouse NP 2.0 recording that has been subjected to mechanical saw-tooth motion [27]. (**C**) The 3D localization images of spikes after Poisson denoising in a thousand second recording are displayed in comparison to the planar localizations provided by Hurwitz et al [11] and the center of mass technique. Poisson denoising the localization point clouds for all three techniques yields a continuous image representation of spiking unit locations and a one thousand frame video representation (one frame for each second of data). (**D**) We apply the existing registration technique [29] on time-binned image representations of data to estimate the amount of z, x, and y motion for all three localization techniques. We show the motion estimate for each localization technique, with and without Poisson denoising. Poisson denoising significantly improves the noise jitter in motion estimation. (**E**) Visualizing z-direction raster plots of the unregistered and registered recordings (after Poisson denoising) shows stabilization of motion effects for all three methods with nominal improvements by our method over others. Green arrows denote areas of the raster plot that have been well stabilized using our localization versus the localization of Hurwitz et al. (**F, H, J, K**) Visualizing the average image after registration using our localization shows significant decrease in image entropy (as a measure of localization "sharpness") over compared methods. (**G**) Additionally, our localization affords the highest average correlation of registered images to the average image and the lowest RMSE. Note that CoM method's high average correlation after registration should be contrasted with its high values prior to registration. Since this localization provides highly blurred images, the average correlation after registration is vacuously high. See supplementary material and video (**video-figure-5.mp4**) for a more detailed comparisons of unregistered/registered frames using the three localization techniques.

methods after Poisson denoising (**Fig. 5E**). This is essentially a maximum **y-** and **x-** projection of registered images, retaining a singleton dimension **z-** to evaluate residual jitter. The registered rasters using localization images from all three methods are qualitatively well stabilized, with nominal improvements afforded by our localization (highlighted by green arrows in (**Fig. 5E**). Evaluating the registration performance using all dimensions shows that our localization, thanks to its three-dimensional spread of features, provides a better reduction of residual motion (**Fig. 5F,G**) and sharper alignment of images (**Fig. 5H,J,K**) compared to the registration using Hurwitz et al. [11] and CoM localization images.

In further detail, we evaluate registration performance using three metrics. First, we compute the average correlation of aligned images to the average image as commonly evaluated in the image registration literature [22]. We contrast this with average correlation before alignment to note the improvement in correlation that alignment provides. Note that the CoM average correlation is high both before and after alignment, denoting that this score is not due to good motion correction but rather to the overall spread of CoM localization that vacuously yields a high correlation to the average image regardless of motion correction. In this metric, our method outperforms both Hurwitz et al. and CoM. We also compute the RMSE of the individual frames to the average frame after registration. Our method also yields the lowest RMSE, showing that individual frames align well to the average frame. Lastly, we evaluate the "sharpness" of the average frame by computing its image entropy. In this regard, our average frame is shown to be qualitatively (**Fig. 5H,J,K**) and quantitatively (**Fig. 5F**) sharper than the results from the compared methods.

## 4    Conclusion

In summary, we provide a simple denoising and point-model triangulation approach to infer three-dimensional source locations from individual spikes in Neuropixels recordings [13, 27]. This localization is shown to facilitate better clustering of distinct neural units as well as enabling better estimation and correction of motion in datasets where the spiking units experience non-stationarities due to probe drift. We also demonstrate the benefits of converting the spiking unit localization point sets into continuous images using Poissonian denoising [18], enabling informative visualizations of the neural source distributions in the vicinity of the Neuropixels probe.

Two open directions are clear for future work. First, as emphasized above, we have used a highly simplified point model to perform localization here. Elaborations of this model should lead to improved accuracy, though robustness and speed of the resulting localizer are also critical for downstream applications and should be balanced accordingly. Second, establishing the accuracy of the proposed methods experimentally will be a challenging but critical next step.

## Broader impact

Neuropixels probes are a recently-developed neurotechnology that enable us to record from hundreds of neurons simultaneously in multiple regions of the brain. Our work will improve scientific conclusions derived from Neuropixels recordings; downstream, we expect related methods to also improve the performance of brain machine interfaces based on extracellular neural recordings.

## Acknowledgments

We thank Nick Steinmetz for collecting and sharing Neuropixels recordings, Olivier Winter and the International Brain Lab for testing our methods, Cole Hurwitz and Matthias Henning for helping us draw comparisons with their localization method, as well as Alessio Buccino and Samuel Garcia for helpful discussions and references. We also thank Cyrille Rossant for providing support for the Datoviz platform. This work was supported by the following grants: Gatsby Charitable Foundation GAT3708, NSF DBI-1707398, NSF 1546296, NIH U19NS104649, NIH U19NS123716, Simons Foundation 543023, Wellcome Trust 209558, and Wellcome Trust 216324.

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
