# Supplementary material: Three-dimensional spike localization and improved motion correction for Neuropixels recordings

**Julien Boussard**\*    **Erdem Varol**\*    **Hyun Dong Lee**    **Nishchal Dethe**    **Liam Paninski**

Department of Statistics and Neuroscience, Center for Theoretical Neuroscience, Grossman Center for the Statistics of Mind, Zuckerman Institute, Columbia University; *-equal contribution

**Supplementary material main contributions:**

1. Details on the localization, motion estimation and registration procedures, as well as a discussion on the importance of our denoising step
2. An illustration and evaluation of the localization method on a synthetic toy dataset
3. Comparison between our 3d localization features and waveforms' shape features
4. Neuropixels 1.0 localization and registration results
5. Videos to illustrate the performance of our improved registration method

## 1 Localization and optimization

### 1.1 Localization and optimization details

The individual waveform neural network denoiser we use to get clean waveforms and un-biased amplitudes is defined in YASS [8]. For each type of probe, the neural network can be trained following instructions at `https://github.com/paninski-lab/yass/wiki/Neural-Networks---Loading-and-Retraining`.

This neural network is an individual waveform denoiser. It denoises the signal on each channel separately. When trained, it expects the input waveform's minimum to be found at a timepoint close to a given value. For example, if trained with waveforms that have their minima around timepoint 41, it will return a clean waveform that takes its minimum close to timepoint 41 as well. It uses this information to remove collisions: If a waveform takes its minimum around 60, it will be automatically treated as a collision and the output will be a waveform taking its minimum around 41. For good accuracy across channels, we upsample and align waveforms before denoising to correct for micro-time shifts between channels. This process removes "close" collisions. However, if a "far away" neuron (for example, localized at a very different z-position) fires simultaneously, the signal will be equal to the sum of the collided waveform and noise, and the denoiser will return the denoised waveform instead of detecting it as a collision. It is important to discard this waveform when computing localization.

To remove these "far away" collisions, we need to first run a de-duplication step (implemented in many spike sorters such as YASS [8] and Kilosort [10]), to get an estimate of the localization and its main channel, giving a set $C_m$ of relevant channels.

For each denoised waveform $w_n$, we find its max channel $\mathrm{mc}_{w_n} \in C_m$ and perform localization using the denoised amplitudes recorded at channels $\{\mathrm{mc}_{w_n} - k/2, ..., \mathrm{mc}_{w_n} + k/2\}$ by minimizing $f(x, y, z, \alpha) = \sum_{c \in C_{w_n}} (\mathrm{ptp}_c - \frac{\alpha}{\sqrt{(x-x_c)^2 + (z-z_c)^2 + y^2}})^2$ over $\{x, y, z, \alpha\}$. We optimize this function

35th Conference on Neural Information Processing Systems (NeurIPS 2021).

**Algorithm 1:** Localization

---

**Input** : $N$ detected events $w_n \in \mathcal{W}$, de-duplicated, each associated with a main channel $\mathrm{mc}_m$ and its surrounding channels $C_m$, the number of waveform-specific channels $K$, an initial value for $y$, $y_{\mathrm{init}}$;

**for** $w_n \in \mathcal{W}$ **do**

    Get $w_n$'s waveform on channels $C_m$;

    Align $w_n$'s waveform on $\mathrm{mc}_m$;

    Denoise $w_n$'s waveform using NN-denoiser [8];

    Find $w_n$'s main channel $\mathrm{mc}_{w_n}$ among $C_m$ using denoised waveforms;

    Get the peak-to-peak amplitudes $\mathrm{ptp}_n$ of $w_n$ denoised waveforms on channels $C_{w_n} = \{\mathrm{mc}_{w_n} - k/2, ..., \mathrm{mc}_{w_n} + k/2\}$;

    Consider $f(x, y, z, \alpha) = \sum_{c \in C_{w_n}} (\mathrm{ptp}_c - \frac{\alpha}{\sqrt{(x-x_c)^2 + (z-z_c)^2 + y^2}})^2$;

    Find an estimate of the global minimizer of $f$, $x_{w_n}, y_{w_n}, z_{w_n}, \alpha_{w_n}$ using least-squares optimization with a center-of-mass initialization ;

**end**

**Output:** Spatial locations $\{x_{w_n}, y_{w_n}, z_{w_n}\}$ and "brightnesses" $\alpha_{w_n}$ feature for every waveform $w_n \in \mathcal{W}$

---

using least-squares optimization after using center-of-mass as a simple quick initialization. The function we optimize is non-convex, but we show in section 1.2. that this optimization method is suited to the task. The whole procedure is summarized in Algorithm 1.

$k$, the number of channels used for localizing denoised waveforms, is the only hyperparameter of our model. Choosing $k$ to be small will lead to "flat" clusters (exhibiting small standard deviation along z-axis and high standard deviation along x-axis), and loss of precision along the z-axis, while choosing a large $k$ will allow small amplitude, noisy channels to be taken into account when localizing, which will spread out the clusters and lead to a loss in precision. The set of "relevant" channels $C_m$ has to be larger than $k$.

### 1.2 Evaluating optimization on a toy dataset

We validated the method on a toy dataset. We reproduced the position of 80 channels, for both NP1.0 and NP2.0, and randomly assigned $\{x, y, z\}$ positions of 1000 neurons drawn from a Gaussian distribution. We then fix $\alpha = 500$ and compute the corresponding amplitudes on each channel using $\mathrm{ptp}_{c,n} = \frac{\alpha}{d_{c,n}}$ where $d_{c,n}$ is the distance of $\{x_n, y_n, z_n\}$ to channel $c$. We use our localization method to infer the corresponding locations, and show that our model can accurately recover $\{x_n, y_n, z_n\}$ locations (**Fig. 1**).

### 1.3 Reliable estimation of spikes amplitudes using the neural network denoiser

To explore the quality of our amplitude estimation using the neural network denoiser described in section **2.2**, we selected 170 clean templates obtained after running YASS [6] on a Neuropixels 2.0 dataset, added background signal taken randomly from the same dataset to get simulated waveforms, and computed amplitudes before and after denoising. **Fig. 2** shows a scatter plot of the true template amplitudes vs. the waveform amplitudes (in blue) and the denoised waveform amplitudes (in red). Denoised amplitudes are much closer to the true amplitudes of the templates, as desired.

### 1.4 Localizations without denoising

To further illustrate the importance of the denoising step on localization, **Fig. 3** (analogous to **Fig. 3** in the main text) shows the inferred features and associated GMM clusters for localizations without denoising. The clusters of locations appear noisier and more spread out. Moreover, we see that many collided spikes belonging to different units are located in single clusters, since there is no way to disambiguate collisions with the amplitude-based localization method.

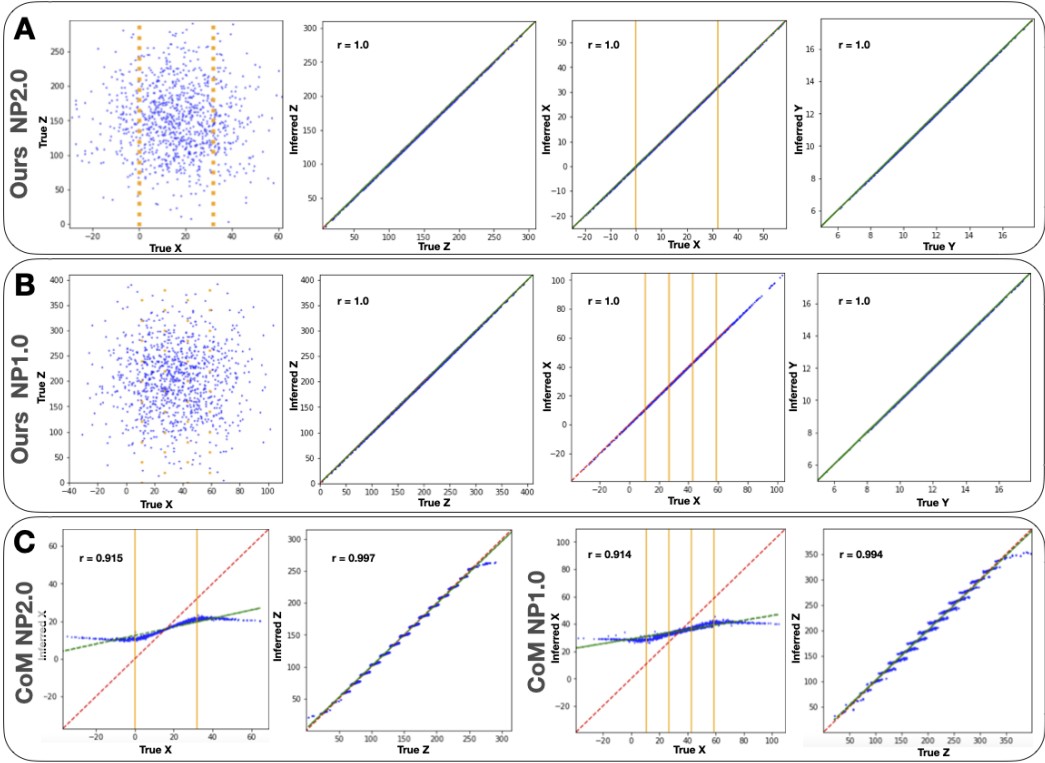

Figure 1: **Simulated toy data localization.** We sampled $\{x, z, y\}$ spike locations from a Gaussian distribution, and computed the amplitudes on each channel using the simple point model. We compare the true locations vs. our model's inferred locations. **(A)** shows results with channels following Neuropixels 2.0 geometry. Left shows the simulated $\{x, z\}$ locations (blue) and the channel positions (orange). The three scatter plots show inferred $\{z, x, y\}$ vs true $\{z, x, y\}$. Dashed red line corresponds to "$x = y$" and the green one to the regression line. Correlation coefficients are reported on the top left of each scatter plot. The orange lines on the third scatter plot indicate the $x$ position of the channels. Panel **(B)** is similar with channels reproducing the geometry of Neuropixels 1.0 probes. Panel **(C)** shows similar scatter plots for the inferred $\{x, z\}$ positions of Center of Mass method, for both Neuropixels 2.0 (left) and Neuropixels 1.0 (right). Our method recovers $\{x, y, z\}$ positions accurately. The Center of Mass method fails at recovering $\{x, z\}$; it localizes inside the convex-hull of the electrode locations, and "shrinks" positions to the center of the probe. The stair-like pattern on the $z$ scatterplots indicate that it tends to localize very close to the channels. We did not compare with Hurwitz et al. method [5] as it does not rely on amplitudes only for localization.

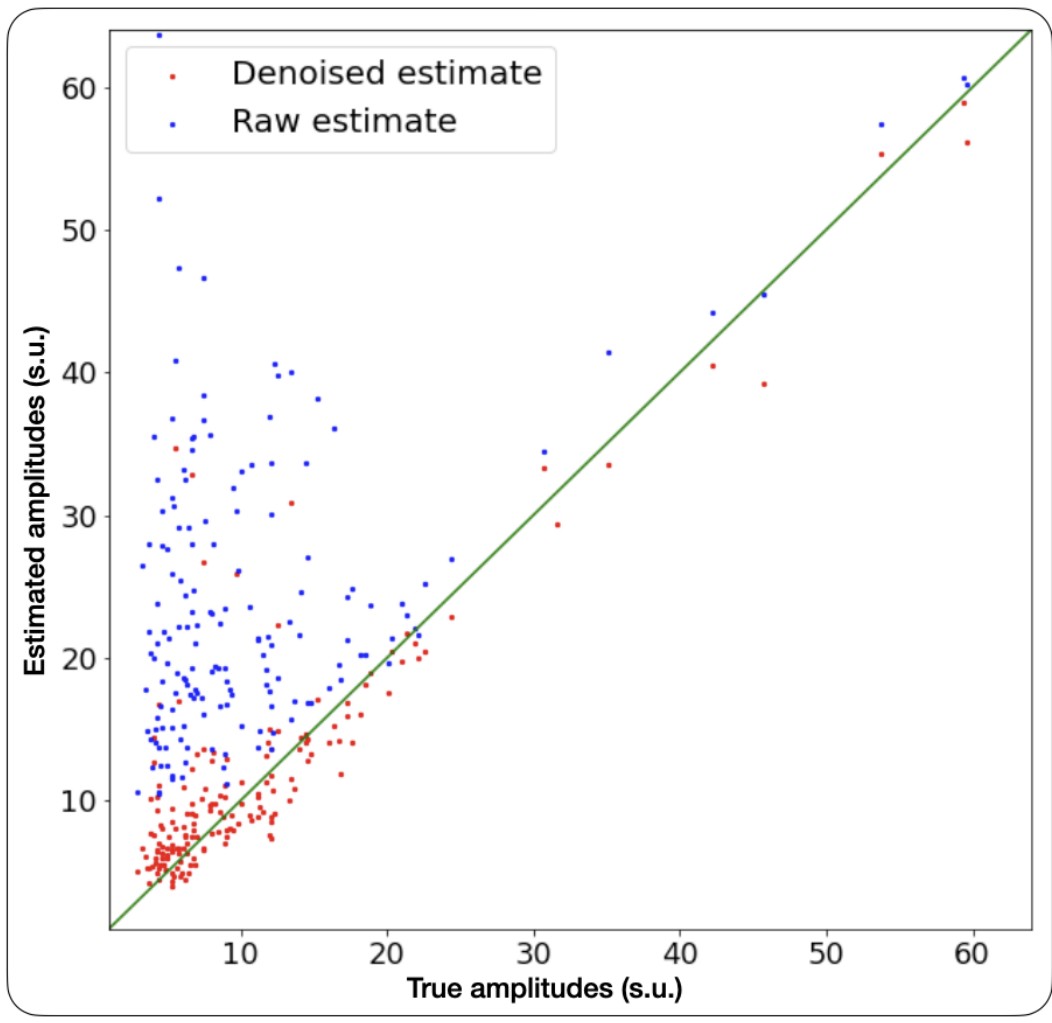

Figure 2: **Denoiser's output waveforms have close to true amplitudes.** Scatter plot of the true template amplitudes vs simulated waveform amplitudes (in blue) and the denoised waveforms amplitudes (in red). (See text for simulation details.) Green line indicates $y = x$. Overall, denoised amplitudes are much closer to the true amplitudes.

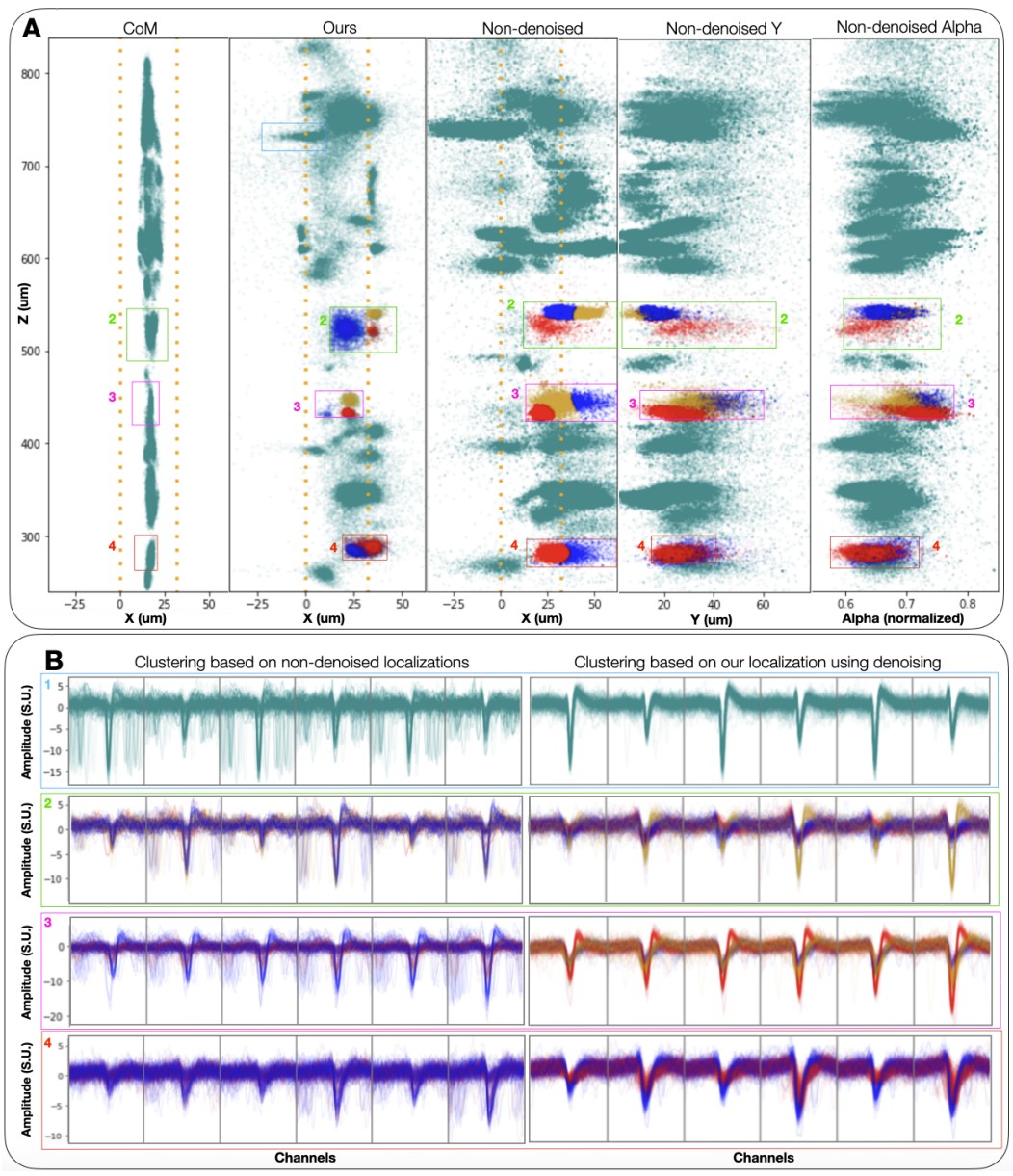

Figure 3: **Denoising lead to sharper clusters and disambiguate collided spikes.** We use the same convention as the main text Fig. 3 in this figure. (**A**) {x, z} locations of spikes detected from 200 seconds of NP 2.0 recording, inferred by the center of mass baseline (left), our method (second to left), and our method without the denoising step (middle). The two scatter plots on the right of the figure represent z vs. y and $\alpha$. The four colored boxes frame spikes that are shown in panel B, and the yellow, blue, and red color of the spikes represent GMM cluster assignments, using {x, z, $\alpha$} as features for our method with and without denoising. The number of clusters has been chosen by hand to reflect the shape of the point clouds in each box. The boxes in each column are not the same size, as we are trying to match spikes localized in the same z-area, which depends on the localization method. (**B**) Waveforms corresponding to each box and cluster, for our method with (right) and without (left) denoising. The many non-centered waveforms in the left column show collided spikes that have not been localized properly without denoising.

## 2 Comparing features

Figure 4 shows a comparison of our localization features $\{x, y, z, \alpha\}$ to the first two principal components of the waveforms, and an additional "spread" feature, equal to the trace of the covariance of the distribution of each spike amplitudes across channels. This feature should be informative of $y$ location, as the maximum amplitude can't help distinguish between a low-amplitude, close spike and a high-amplitude, far away spike.

Some clusters are better separated by $\alpha$ than any other features, indicating that it contains information about the shape or location of the waveforms that is not contained by other single features.

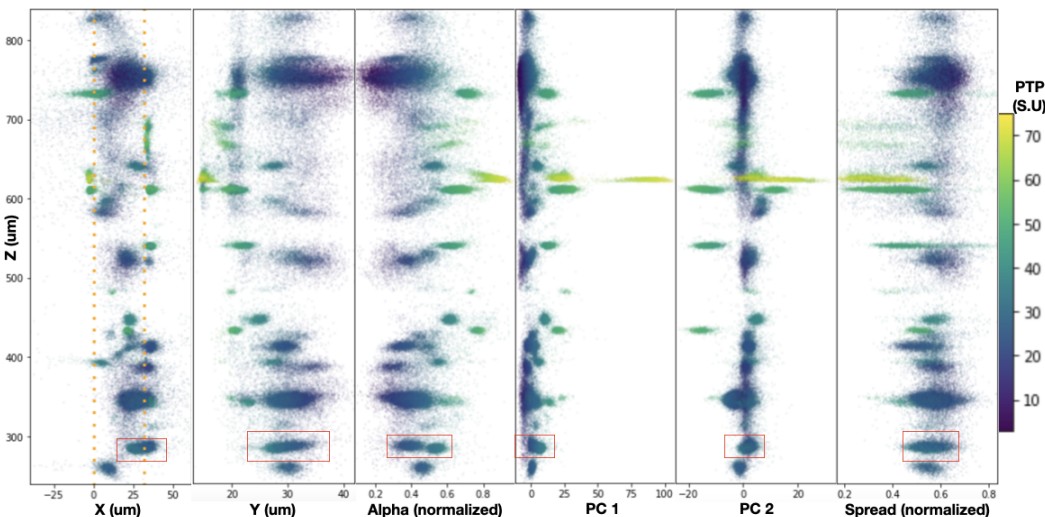

Figure 4: **Comparing localization and shape features.** Scatter plots showing our inferred $\{x, z\}$ locations of spikes detected from 200 seconds of NP 2.0 recording (left), $\{y, z\}$ locations (second to left), $\{\alpha, z\}$ (third to left), the first two Principal Components of the waveforms vs. $z$ (fourth and fifth to left), and the spread of each waveform vs. $z$ (right). Spikes are colored by maximum peak-to-peak amplitude. The spread of each waveform is calculated as the covariance of the distribution of the denoised amplitude over the set of selected channels. We expect spread to be informative about $y$, as a far away high-amplitude spike will be seen on multiple channels with low detected amplitude, giving high spread, whereas a small or high amplitude spike close to the channels will correspond to a low spread value. $Y$ is determined by the amplitudes. The point clouds inside the red box (around depth 300) are only separated by $\alpha$, and not by the Principal Components of the waveforms, the spread or the maximum amplitude, suggesting that $\alpha$ contains additional information useful for clustering. This figure shows scatter plots corresponding to the same spikes as the main text Figure 3, and the red box corresponds to Box 4.

# 3 Comparison of image denoising techniques

Due to the limitations of the point neuron model and the sparse number of observations, the localization methods provide a partial picture of the spatial layout of spiking units. To generate a dense localization images for image based registration, we utilize three different denoising techniques: 1) Gaussian smoothing, Poisson denoising [9] (ours), and Deep interpolation (DI) [7].

- Gaussian smoothing involves blurring the localization images by a Gaussian filter to infill gaps. Here we used a kernel size of $5\mu$ m.

- DI is a neural-network based denoising algorithm that takes noisy samples from the original raw data as inputs to train a spatio-temporal nonlinear interpolation model. We applied deep interpolation to a small patch of NP 2.0 raster image (depth $70\text{-}582\mu$m and time 1-512s), with one of the network architectures provided by the authors ("unet_single_256"). The network is trained with 5 steps per epoch (total of 7 epochs), with batch size of 1 and "pre_post_frame" set to 1.

- Poisson denoising models the observed localization image as having been corrupted by Poisson salt-and-pepper noise, with the likelihood of observing a spike proportional to its amplitude. Since this process models the noise variance to be proportional to its mean, we first apply the Anscombe transformation [1] to stabilize noise variance prior to denoising the transformed localization image by BM3D [4] or non-local-means [3]. After the transformed image is "denoised" we apply the inverse Anscombe transformation to yield the "Poisson denoised" localization image.

The three approaches to image denoising have slightly different effects on downstream motion estimation. Figure 5 shows the comparison of the probe displacements for the first 512 seconds, estimated by localization images denoised by Gaussian filtering, DI, and Poisson denoising. The estimates are similar, but there is a drastic difference in the run-time (178 for Deep Interpolation vs. 0.27 seconds for Poisson denoising), suggesting that our Poisson denoiser is both effective and efficient. Furthermore, the motion estimate curves yielded by DI outputs of localization images shows dampened peak-to-trough amplitudes of motion in the simulated NP 2.0 datasets, indicating that DI may be slightly over-smoothing the images, obscuring fine details that may be useful for precise motion estimation.

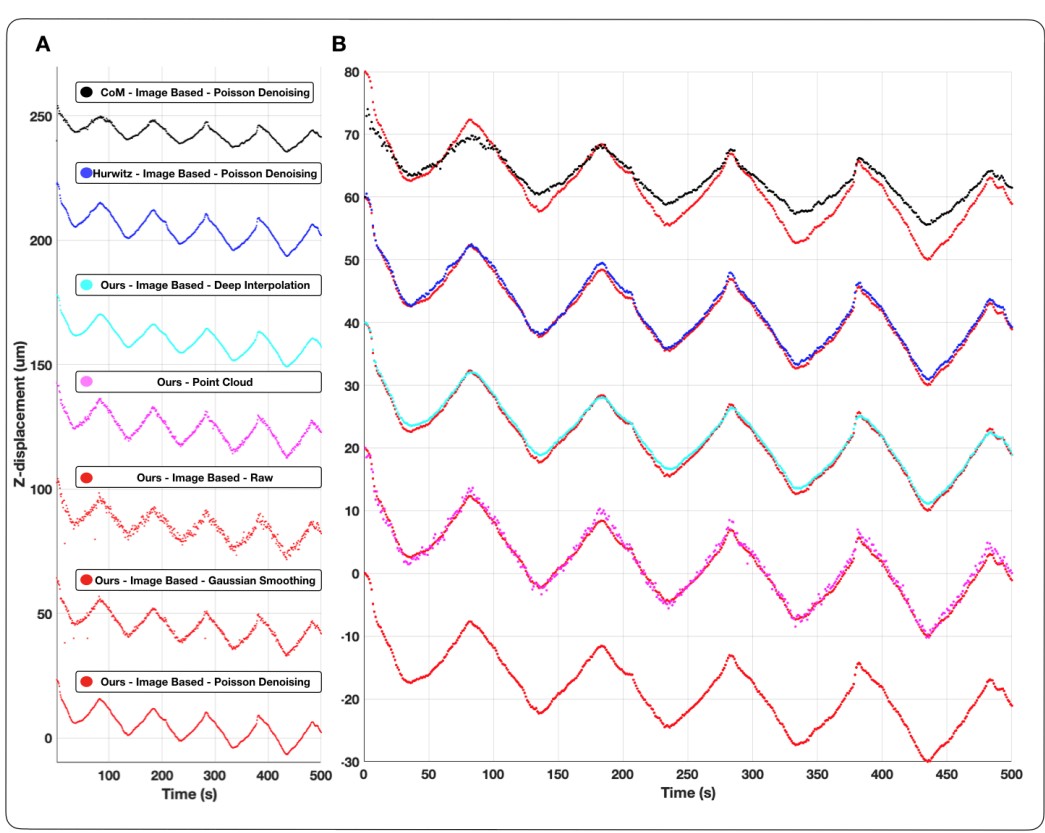

Figure 5: **Comparing the effects of different image denoising techniques for motion estimation A:** The displacement estimates obtained from our localization images without denoising as well as Poisson denoising, Gaussian smoothing, and Deep Interpolation denoising, compared with the displacement estimates obtained using Center of Mass localization and Hurwitz method of localization with Poisson denoising. **B:** The displacement estimates are superimposed on top of the estimates from our localization method + Poisson denoising to examine subtle differences between the methods. Displacement estimates using our localization method yield similar outputs whether we use image based registration or point cloud based registration. Our localization with Poisson denoising yields slightly "peakier" displacement estimates versus denoising with Deep Interpolation. Gaussian blurring yields noisier estimates of displacement than the other denoising methods.

# 4 Additional details on point-cloud registration

Algorithms 2 and 3 outline the detailed steps of the point-cloud registration technique (in the rigid case, for simplicity), with Figure 6 providing an illustration.

---

**Algorithm 2:** Point-cloud registration

---

**Input** : Localization coordinates, amplitudes and spikes times $\{x_{w_n}, z_{w_n}, y_{w_n}, ptp_{mc_{w_n}}, t_{w_n}\}$
 for $w_n \in \mathcal{W}$; threshold distance $d_{outlier}$ for removing outliers; the maximum merge distance
 $d_{merge}$ for agglomerative clustering; grid search range $G$ and stride $stride$;
// Preprocessing
**for** $T = 1, 2 \ldots, \max(t_{w_n})$ **do**
 | Generate point cloud $\mathcal{PC}_T(x, y, z, ptp, n)$ from $w_n \in \mathcal{W}, T \leq t_{w_n} < T + 1$:
 |     Remove outliers: remove $w_n$ if its average distance to $k$ nearest neighbors is greater
 |   than $d_{outlier}$
 |     Run agglomerative clustering with $d_{merge}$: Recursively merge pairs of clusters -
 |   $\{x, y, z, ptp\}$ are averaged when merged, and $n$ represents the number of spikes in the
 |   cluster
**end**
// Run modified iterative closest point
**for** $T = 1, 2 \ldots, \max(t_{w_n})$ **do**
 | **for** $T' = 1, 2 \ldots, \max(t_{w_n})$ **do**
 | | // Initialize with grid search
 | | Find $\Delta z_{T,T'} \in range(-G, G, stride)$ such that $computeLoss(\mathcal{PC}_T, \mathcal{PC}_{T'}, \Delta z_{T,T'})$ is
 | |   minimized.
 | | Run iterative closest point [2] to update $\Delta z_{T,T'}$ such that
 | |   $computeLoss(\mathcal{PC}_T, \mathcal{PC}_{T'}, \Delta z_{T,T'})$ is minimized, with maximum iteration $maxIter$
 | **end**
**end**
Estimate global $z$ positioning of each time bin of data using method in [12]: $p_z(T)$;
**Output:** Motion estimate for each one-second of data $p_z(T)$.

---

---

**Algorithm 3:** $computeLoss$

---

**Input** : A pair of point clouds $\mathcal{PC}_T(x, y, z, ptp, n)$, $\mathcal{PC}_{T'}(x, y, z, ptp, n)$; $z$-displacement
 $\Delta z_{T,T'}$; a threshold distance $d_{threshold}$ for filtering out pairs of points
Shift $\mathcal{PC}_{T'}(x, y, z, ptp, n)$ in $z$-direction by $\Delta z_{T,T'}$
// Distance from $\mathcal{PC}_T(x, y, z, ptp, n)$ to $\mathcal{PC}_{T'}(x, y, z, ptp, n)$
Update $L$: Add average L2-distance (in $x$, $y$, and $z$ dimensions) from points in
 $\mathcal{PC}_T(x, y, z, ptp, n)$ to their nearest neighbors in $\mathcal{PC}_{T'}(x, y, z, ptp, n)$, weighted by $ptp$ and $n$,
 after discarding pairs of points with L2-distance greater than $d_{threshold}$
// Filter $\mathcal{PC}_{T'}(x, y, z, ptp, n)$
Mask out points in $\mathcal{PC}_{T'}(x, y, z, ptp, n)$ that are not selected as nearest neigbors of
 $\mathcal{PC}_T(x, y, z, ptp, n)$ in the previous step
// Distance from $\mathcal{PC}_{T'}(x, y, z, ptp, n)$ to $\mathcal{PC}_T(x, y, z, ptp, n)$
Update $L$: Add average L2-distance (in $x$, $y$, and $z$ dimensions) from points in
 $\mathcal{PC}_{T'}(x, y, z, ptp, n)$ to their nearest neighbors in $\mathcal{PC}_T(x, y, z, ptp, n)$, weighted by $ptp$ and $n$,
 after discarding pairs of points with L2-distance greater than $d_{threshold}$
**Output:** Loss $L$.

---

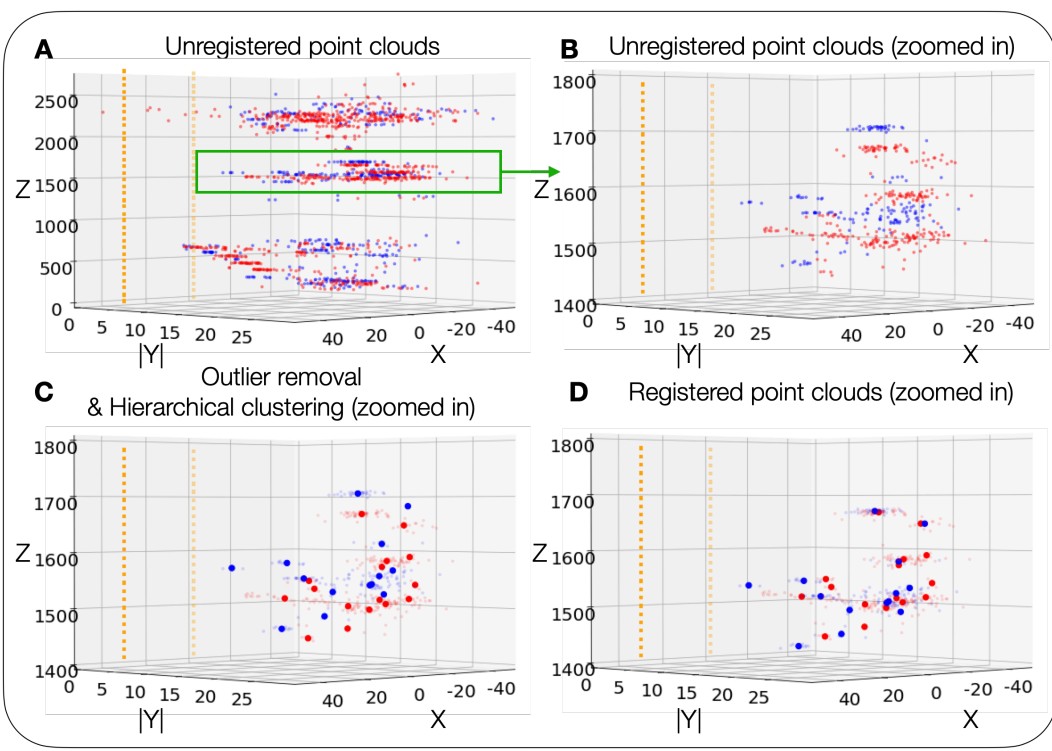

Figure 6: **Visualization of point-cloud registration on NP 2.0 recording.** Visualization of unregistered point clouds (**A**), unregistered point clouds zoomed in to depth $1400$ - $1800\mu$m (**B**), compressed (i.e. outlier removal & hierarchical clustering) point clouds (**C**), and registered compressed point clouds (**D**). Colors represent which second of data each spike is from (red: 1st second, blue: 850th second). Orange squares represent the recording sites of the NP 2.0 probe. Smaller dots represent the original spikes, and larger ones represent the means of the hierarchical clusters. Here we register the blue point cloud to the red one by shifting $-33.04\mu$m in the $z$-direction, and we see that the two point clouds overlap to a greater extent after registration.

# 5 Neuropixels 1.0 results

## 5.1 Spike localization

To demonstrate that the localization model is robust to different types of probes, we show improvements over previous localization methods on a Neuropixels 1.0 dataset. **Fig. 7** shows locations inferred by Center of Mass, Hurwitz et al. [5], and our method, and corresponding clusters obtained using Gaussian Mixture Model on the location features. For the Hurwitz et al. method, channels within 50 $\mu m$ from the main channel are included (10 observed amplitudes) and amplitude jitter is set to 0 $\mu V$.

## 5.2 Motion estimation and registration

We evaluate registration performance to demonstrate that motion estimation is robust to the localizations that arise from Neuropixels 1.0 probes geometry. The quantitative and qualitative results are shown in **Fig. 8**.

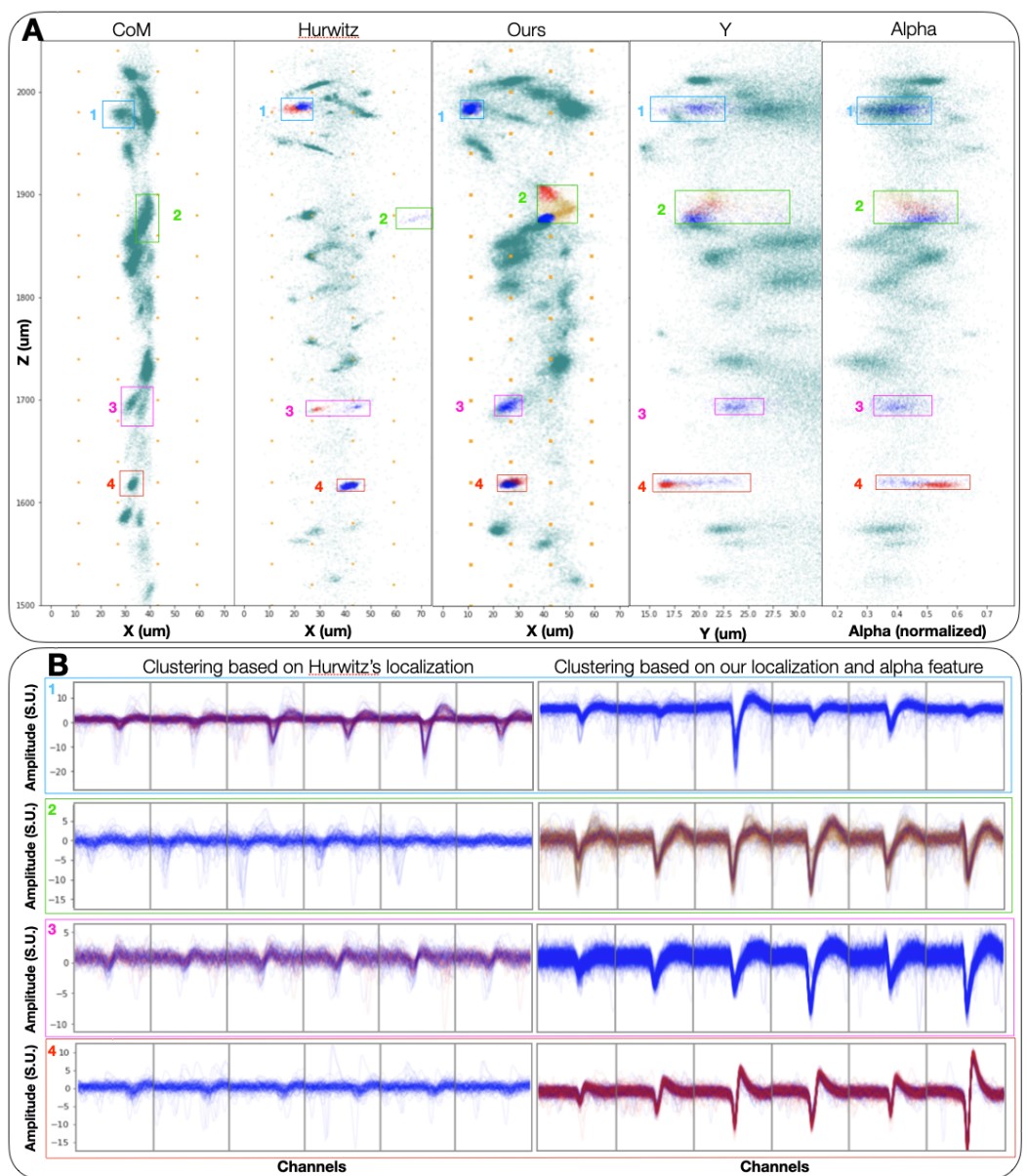

Figure 7: **Inferred 3D spatial features yield improvements in Neuropixels 1.0 waveform clustering (analogous to figure 3 in main text)** (**A**) shows center of mass, Hurwitz et al. and our localization results, with added features $Y, \alpha$ for our method. Spikes in each box correspond to the waveforms in panel (**B**), with blue, red and yellow colors corresponding to Gaussian Mixture Model clustering with the number of components reflecting the cloud points shape. The many non-centered waveforms in the left column of Panel (B) show collided spikes that have not been localized properly by the Hurwitz et al. method, which often spatially separates similar waveforms, while failing to isolate different units. On the other hand, our location-based clusters correspond to similar units for boxes 1 and 3, without corruption from poorly localized collided spikes. Nonetheless, some units here show signs of oversplitting (e.g., boxes 2 and 4), indicating room for potential further improvement.

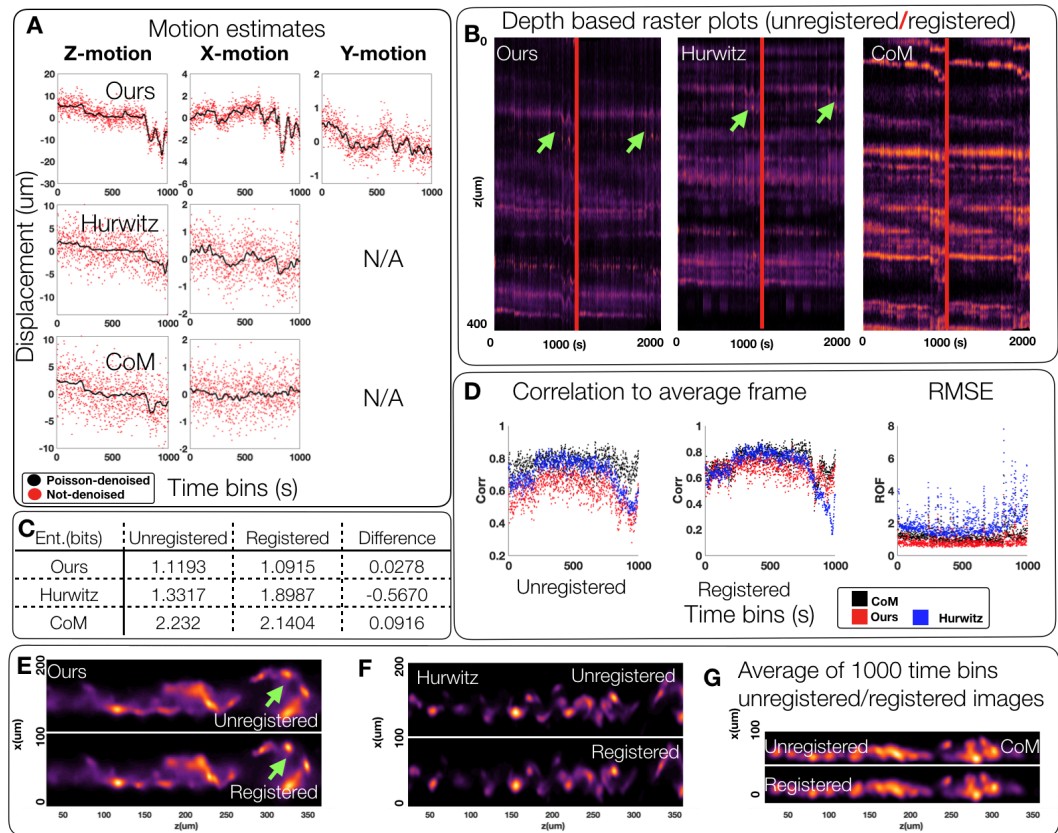

Figure 8: **Improved localization enables better motion correction and registration of Neuropixels 1.0 data (analogous to figure 5 in main text)** (**A**) We apply the existing registration technique [12] on time-binned image representations of data to estimate the amount of z, x, and y motion for all three localization techniques. We show the motion estimate for each localization technique, with and without Poisson denoising. Poisson denoising significantly improves the noise jitter in motion estimation. (**B**) Visualizing z-direction raster plots of the unregistered and registered recordings (after Poisson denoising) shows stabilization of motion effects for all three methods with nominal improvements by our method over others. Green arrows denote areas of the raster plot that have been well stabilized using our localization versus the localization of Hurwitz et al. (**C, E, F, G**) Visualizing the average image after registration using our localization shows significant decrease in image entropy (as a measure of localization "sharpness") over compared methods. (**D**) Additionally, our localization affords the highest average correlation of registered images to the average image and the lowest RMSE. Note that CoM method's high average correlation after registration should be contrasted with its high values prior to registration. Since this localization provides highly blurred images, the average correlation after registration is vacuously high.

# 6 Supplementary videos

## 6.1 Supplementary Datoviz video for figure 1 in main text (Web link: video-figure-1.mp4)

Datoviz [11] is a high-performance interactive data visualization library that we use to visualize our localizations and spike sorting output along the probe, and inspect the clusters in 3-d. In this example video, we are showing four different panels. The first one displays the spikes' 3-d locations $\{\mathbf{x}, \mathbf{y}, \mathbf{z}\}$ along the probe, colored by post spike-sorting clusters. The second panel shows the same features, colored by maximum amplitude. The third one shows the same features colored by $\alpha$ feature, while the last one displays $\{\mathbf{x}, \mathbf{z}, \alpha\}$ along the probe, colored by maximum amplitude.

This interactive plot allows us to scroll along the probe, zoom to inspect clusters (do the spike-sorting units have well-defined clusters, do they correspond to two different clusters, is one cluster separated in two different units?), aggregate spikes in time, and look at different time points in the recordings. The latter provides a good way to visualize the 3-d displacement of the probe.

## 6.2 Supplementary video for figure 4 in the main text (Web link: video-figure-4.mp4)

This video shows the zoomed regions of the Neuropixels probe that correspond to the anatomical regions of cortex, hippocampus and thalamus. Screenshot of this video is shown in figure 9. In each frame top left panel shows the sparse localization image of detected spikes and their spatial positions after having been projected along the depth (y) axis. Top right panel is analogous to top left panel but shows the projection along the horizontal axis (x). Middle panels show the localization images after having gone through Poisson denoising [9]. Bottom panels show the Poisson denoised images after motion estimation and registration.

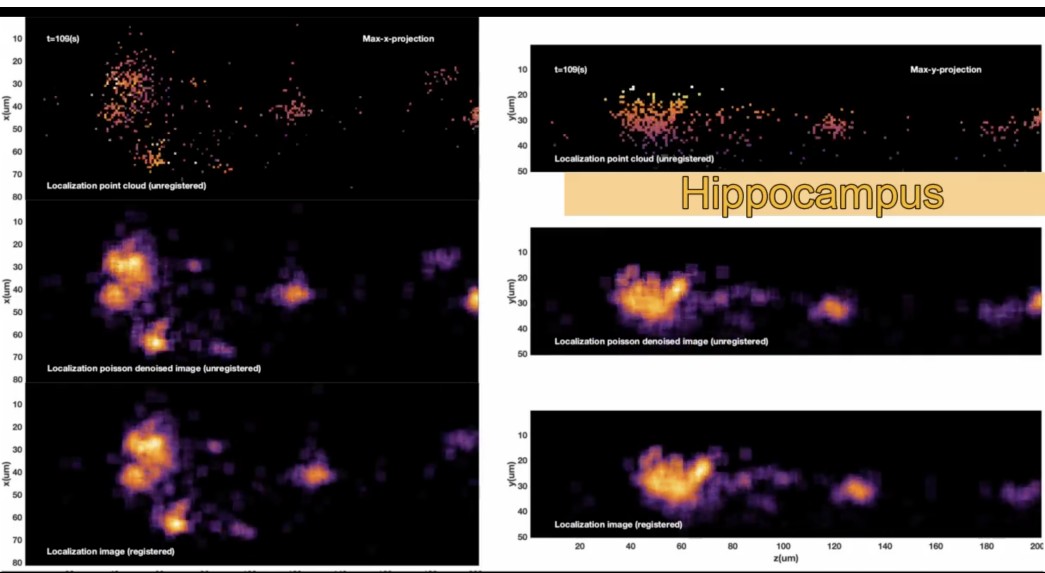

Figure 9: **Video representation of Poisson denoising, and motion estimation in zoomed regions of cortex, hippocampus, and thalamus in Neuropixels 2.0 data**. This is a screenshot of supplementary video for figure 4 in main text showing a frame that corresponds to the hippocampal zoomed region. In each frame top left panel shows the sparse localization image of detected spikes and their spatial positions after having been projected along the depth (y) axis. Top right panel is analogous to top left panel but shows the projection along the horizontal axis (x). Middle panels show the localization images after having gone through Poisson denoising [9]. Bottom panels show the Poisson denoised images after motion estimation and registration. Note that in all anatomical regions, the motion has been mitigated in the registered panel.

## 6.3 Supplementary video for figure 5 in main text (Web link: video-figure-5.mp4)

This video compares the localization images and the registration performance using all three compared methods (ours, Hurwitz et al. and CoM) in both Neuropixels 1.0 and 2.0 datasets. Screenshot of this video is shown in figure 10.

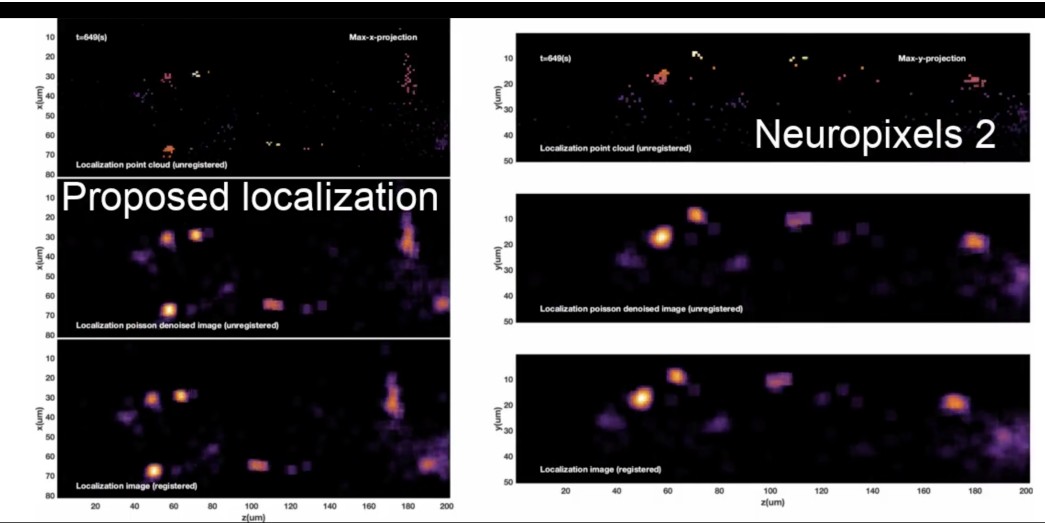

Figure 10: **Video representation of Poisson denoising, and motion estimation using compared localization methods in both Neuropixels 1.0 and 2.0 data**. This is a screenshot of supplementary video for figure 5 in main text showing a frame that corresponds to the cortical zoomed region using localization and registration guided by our technique in Neuropixels 2.0 data. Subsequent frames show comparisons with Hurwitz et al. and CoM in both Neuropixels 1.0 and 2.0 data. In each frame top left panel shows the sparse localization image of detected spikes and their spatial positions after having been projected along the depth (y) axis. Top right panel is analogous to top left panel but shows the projection along the horizontal axis (x). Middle panels show the localization images after having gone through Poisson denoising [9]. Bottom panels show the Poisson denoised images after motion estimation and registration. Note that the periodic horizontal motion visible in the middle (unregistered) panels is mitigated in the bottom panels after registration, demonstrating that the estimated motion corresponds well to the underlying motion. Also note that our proposed method yields a more stabilized registration. Since the CoM method compresses localization to a narrow strip, features that could be used to guide a fine resolution registration are lost, causing an underestimation of motion.

# 7 Datasets

For reproducibility of our experiments, datasets used in our paper can be found here : `https://github.com/flatironinstitute/neuropixels-data-sep-2020/blob/master/doc/cortexlab1.md`