# OpenReview forum: "Three-dimensional spike localization and improved motion correction for Neuropixels recordings"
_NeurIPS.cc/2021/Conference — NeurIPS 2021 Poster_

### Official Review · Reviewer_PPLm · 2021-06-30

**Rating:** 5
**Confidence:** 4

**Summary:**

The authors describe a pipeline for processing recordings from neuropixel probes which are the current state of the art in in-vivo recordings from single neurons in the brain.

**Ethical Concerns:**

none.

**Limitations And Societal Impact:**

Limitations were only superficially discussed, but I don't see negative societal impact of this work. The authors could add a discussion on potential improvements.

**Main Review:**

The authors describe a pipeline for processing recordings from neuropixel probes which are the current state of the art in in-vivo recordings from single neurons in the brain. Their pipeline consists of (a) a point neuron model for spike localization in 3D, (b) a denoising step for the waveforms and the resulting localizations, (c) motion registration and (d) spatial clustering. This is an interesting approach, but the way the paper is presented this seems more like a description and validation of a toolbox than a machine learning application paper. Also, step (b) is simply reused from another paper (which is totally fine, but hard to sell as a contribution then).

An additional issue with the paper is that in many instances (e.g. the denoising step for the waveforms and the images) the result is not really validated and only very briefly explained. Why is it “correct” to remove all the events in line 2 of the figure to the right? I suppose these are collisions, but isn’t it dangerous to remove them altogether? Weren’t real spike the cause of these as well?

For the image denoising, how does it lead to different results than simply smoothing the image with a Gaussian kernel? The authors state in line 98 a variety of methods they actually could use for denoising, which was it? Does the choice of method influence the result? Finally, I am not convinced taking the square root of count data and applying Gaussian denoising should be called Poisson denoising.

The remaining results look very promising, yet extremely dense for such a conference paper. Therefore, I would recommend a more extended format at a neuroinformatics/comp neuro journal venue. After considering the rebuttal and the discussion, I am still not convinced that this form of paper with extensive supplement is ideal for NeurIPS, but can see the point that it is within scope.


**Time Spent Reviewing:**

1

---

> ### Author Response · Authors · 2021-08-10
> **Response to reviewer PPLm**
>
> We thank the reviewer for praising the completeness and density of our work and appreciating the promising nature of our results. However, we respectfully disagree that our paper is not appropriate for Neurips. As stated earlier in response to R1, Neurips has a long and impressive history of publishing strong neuroscience papers --- indeed, the Neurips 2021 Call for Papers specifically solicits applied neuroscience papers --- and specifically, many highly influential spike sorting papers appeared at Neurips (including the original Kilosort (Pachitariu et al. 2016) and YASS (Lee et al. 2017) papers, and the paper by Hurwitz et al. 2019 on spike localization that our work builds upon). Our paper fits perfectly within this tradition, and therefore we ask R4 to reconsider their score.
>
> We have additionally addressed all of the technical comments that the reviewer raised.
>
> On the clarity of contributions:
>
> We have not claimed novelty on the denoising steps that we utilize in the paper but merely use them for the errors that are incurred due to the noisy nature of NP recordings and the inevitable model mismatch our point neuron model experiences. The major novelty of our contribution here is to apply waveform denoising to significantly improve localization and registration performance. We will clarify in the main contributions section the applied nature of our methods in order to avoid confusion by readers.
>
> On the validity of NN denoising for waveforms:
>
> We have performed additional experiments on performing localization without doing NN denoising. These experiments have produced a result similar to figure 3 that we will add in the supplementary material.
> Takeaways :
> Overall localizations are noisier (x,y,z clusters appear more spread out).
> Localization induces clusters that are specific to collisions, i.e., for one neuron, some of its spikes will be collided with the same neuron. The amplitudes on collided channels will correspond to the greater neuron amplitude or sum, and we’ll see the original amplitudes of both spikes on some channels. Localizations of these spikes will lead to another cluster that does not correspond to any of the original neurons.
>
> We also want to clarify that when we “remove” collisions, we don’t throw them away. We expect the spike train to detect the two collided spikes, and our goal is to localize them independently of each other. Thus, when evaluating the amplitudes of each collided spike on the different channels, we need to “remove” the other collided spike so that it is localized similarly to the non-collided spikes of the same neuron.
>
> On the validity of Poisson image denoising:
>
> The nature by which we detect spiking events closely resembles a Poisson process due to the point-process nature of the spiking data. Thus employing Poisson image denoising techniques is a natural choice for our spike rasters that we aim to denoise prior to registration. However, we value the suggestion proposed by the reviewer due to its potential computational simplicity. Thus, we have Gaussian filtered the localization images with several kernels (2um--10um) and performed registration. This result has yielded motion estimates that are marginally “over-smoothed” compared to the estimates obtained from Poisson denoised rasters. We speculate that this is due to the inability of Gaussian filtering to reduce salt and pepper noise present in the rasters, which the internal BM3D or non-local means routine of Poisson image denoising is able to handle (Makitalo, Foi, 2010). We have prepared registration comparison figures for Gaussian filtered vs. Poisson denoised rasters in the supplementary material of the revision and have discussed this finding in the main text results.

---

### Official Review · Reviewer_2yf2 · 2021-07-16

**Rating:** 8
**Confidence:** 4

**Summary:**

In “Three-dimensional spike localization and improved motion correction for Neuropixels recordings”, the authors describe a pipeline for denoising and localizing spikes from dense extracellular recordings of neurons. Specifically, recordings are denoised using a previously published neural network based technique, then 3D locations are inferred using a straightforward point-source model. Finally, spikes are localized using Poisson image denoising.

**Ethical Concerns:**

No ethical concerns.

**Limitations And Societal Impact:**

Yes, no issues here.

**Main Review:**

The manuscript is well-written, and I found the methods easy to follow and design decision are reasonable and justified. This paper in its current form is well-above the acceptance threshold for NeurIPS, and I only have a few comments:

Major:

1)	In the comparison with Hurwitz’s method, my understanding is that the authors do not use any spike amplitude features (that is just x and z are used for Hurwitz). This must be added for a more apples-to-apples comparison between the two methods. Even if it is noisy, the authors should add peak-to-peak estimates to Hurwitz’s feature set.
2)	Re: clustering it would be useful to know how clustering using x, y, and alpha compare with clustering using typical waveform features (e.g. PCA decomposition of the time-course). Directly demonstrating that including position information aids unsupervised clustering would be useful here. Would, for example, someone using Kilosort want to add these position features for clustering neurons?

Minor:

1)	Does the neural network denoising used by the authors induce any distortion of the spike waveforms in these recordings?


**Time Spent Reviewing:**

4

---

> ### Author Response · Authors · 2021-08-10
> **Response to reviewer 2yf2**
>
> We thank the reviewer for the overall high praise of the paper and the constructive comments. Below we address the comments and suggestions one by one:
>
> On apples to apples comparisons with Hurwitz et al. method:
>
> We have indeed used peak-to-peak amplitudes in our run of the Hurwitz et al. method upon consulting with the authors of the paper. The current results in the manuscript reflect that. We will make it more explicit in the results section on the specifics of how the Hurwitz et al. method was run.
>
> On the utility of position information on unsupervised clustering:
>
> We thank the reviewer for raising this excellent point, as this is one of the central theses of our paper -- localization is an important feature for any spike sorting algorithm downstream. To demonstrate this, we have focused on the subsets of spike clusters shown in boxes in Figure 3. We have shown that for these spikes, our localization induces clusters that correspond to a single unit and separate different units correctly. We argue that using these features instead of the usual center of mass positions will lead to improved spike sorting accuracy. We show that using localization + PCA features usually improves clustering stability by measuring the Adjusted Rand Index (ARI) of cluster agreement over randomized re-runs. Below is a text box of the comparisons we have executed for this response that we will include in the revised manuscript:
>
> ```
>               ARI |  Box 2 | Box 3  | Box 4  |
>           Our x,z |   0.98 |  0.79  |  0.74  |
>  Our x, z + 5 PCs |   0.86 |  0.68  |  0.61  |
>  Our x, z + 2 PCs |   0.83 |  0.44  |  0.76  |
>             5 PCs |   0.3  |  0.74  |  0.46  |
>             2 PCs |   0.5  |  0.54  |  0.95  |
>  CoM x, z + 5 PCs |   0.35 |  0.79  |  0.51  |
>  CoM x, z + 2 PCs |   0.89 |  0.36  |  0.87  |
> ```
> CoM abbreviates “center of mass” location estimate here. The ARI values were obtained by running GMM on the x, z and PC features of the spikes inside each box. However, GMM might not be the best clustering algorithm here. Spike sorters such as Kilosort or YASS use more sophisticated clustering algorithms. We hope that inputting the new x, z localization features as well as the additional features y and alpha to the desired clustering algorithm will improve overall accuracy of spike sorting; we will investigate this question further in future work.
>
> On whether NN denoising introduces any distortions of the waveforms:
>
> As there is no ground truth for waveforms, it is hard to evaluate quantitatively the performance of the NN denoiser beyond the simulations we use to train the denoiser. However, looking at raw vs. denoised waveforms, the denoiser seems to output a reasonable estimate of the waveforms. YASS (Lee et al., 2020) Figure 3 displays more examples of neural network performance. For the sake of localization, we need the NN denoiser to remove collisions and provide a good estimate of the amplitudes (not the shapes) of the waveforms; qualitatively, the NN performs well at this task, according to visual inspection of many examples. Quantitatively, when denoising simulated data corresponding to templates + white noise, the denoised PTP is much closer to the true ptp than the raw ptp. We will show scatter plots illustrating this effect in the revised supplementary material.
>
> We have performed additional experiments on performing localization without doing NN denoising. These experiments have produced a result similar to figure 3 that we will add in the supplementary material. They show that without denoising, overall localizations are noisier and localization induces clusters corresponding to collisions (see the response to R4).

---

### Official Review · Reviewer_vagG · 2021-07-16

**Rating:** 10
**Confidence:** 4

**Summary:**

This paper proposes a novel spike sorting algorithm, specifically for dense arrays such as the NeuroPixels series of electrodes. This is an extensive and detailed report. The comparisons to other approaches are up-to-date and thorough. They present some nice analysis of electrode drift, showing that the new algorithm performs well. It is well written, clear, and includes extensive figures to illustrate the work and results. This is an excellent piece of work.

**Ethical Concerns:**

None.

**Limitations And Societal Impact:**

No concerns.

**Main Review:**

They use a point-source model for neurons, then denoise and supress collisions, and finally do triangulation in 3D. This apparently improves clustering fidelity. They also track electrode drift and their algorithm offers excellent performance for that aspect as well.

This work can have impact on both fundamental science, as well as BCI work-- although the latter often disregards spike sorting.

It would be helpful if the report could show some spike raster data comparing their approach and that of Hurwitz and CoM. For example, in Fig 2 there are some clusters that were split in their algorithm and not with Hurwitz et al.'s algorithm. What do those spike rasters look like? Sometimes good spike sorting is evident when aligned to a stimulus or task parameter, e.g., Fig 3 in https://onlinelibrary.wiley.com/doi/epdf/10.1111/joa.12228

The main limitation is that there is no ground truth data, or calibration data. How can we say that this algorithm is better when we don't know what the truth is? This is a general problem for the field, and not a specific problem for this paper, but it does make it awkward to strongly advocate for a particular spike sorting method. Still, I am enthusiastic about this work and strongly support it.

"crisper estimates of neural clusters over time" - This is an unclear, qualitative way to say it. Please use more precise, quantitative language. Perhaps: "more stable unit localization over time, as measured by image entropy"? I defer to the authors, and I will not insist on this, but I would like to encourse the use of more precise, quantitive language.

The model (a point neuron model) is rather simple, which is maybe good, but where does it break down? Spike signals can emerge from different compartments of a neuron. Spike amplitude can be affected by not only distance, but also geometry (e.g., angle of an axon relative to the electrode) and intervening tissue (e.g., glia in front of an electrode). I guess those would just get folded in, and not cause problems as long as they are stable in time-- degrading the precision of the localization, but still aiding the clustering/sorting. I would appreciate the authors commenting on that.

Purely curiosity: The 3-step Poisson denoising is interesting, and seems to work well. I wonder what the author(s) think about the DeepInterpolation work. https://github.com/AllenInstitute/deepinterpolation Have they tried to replace their denoising with this approach and examined performance?

**Time Spent Reviewing:**

2

---

> ### Author Response · Authors · 2021-08-10
> **Response to reviewer vagG**
>
> We thank the reviewer for the thorough review of our paper and are humbled by the highest possible score assigned to us. Below we address the specific technical concerns raised by the reviewer.
>
> On showing the spike rasters produced by our method as well as Hurwitz et al. and CoM methods:
>
> We have already included raster plots before and after motion correction using our method / Hurwitz et al. and CoM in figure 5E. This raster plot shows unsorted spikes arranged in their depth localizations, colored by their amplitudes. In order to not be confounded by spike sorting algorithms’ internal biases and not come off as a spike sorting paper, we decided to evaluate our localizations using unsorted spikes.
>
> However, we agree that it may be informative to see spike localizations colored by clusters to see how units might be oversplit/undersplit due to the quality of localizations. For the supplementary material of the revision, we have prepared a spike sorted raster plot of the compared localization methods using Kilosort/YASS.
>
> Also, the current raster plot in figure 5E probably deserves a larger presentation to appreciate the subtle differences in localizations and we will provide a larger version in the supplementary material. We will also emphasize this in the caption and results to enable the reader to find this subfigure more easily.
>
> On the lack of calibration data:
>
> We agree that the lack of realistic calibration data for NP data prohibits an in-depth evaluation of the performance of many methods, not limited to localization but spike sorting as well. We have made a note of this in the discussion section in the revision. To our knowledge, it is necessary to evaluate the quality of spike sorting by directly looking at the waveforms included in each cluster. This is why we exhibit the waveforms corresponding to different location-induced clusters in Figure 3 to show that the location clusters indeed seem to correspond to neuronal clusters. Moreover, we hope that our localization method will help improve visualization and evaluation of the spike train, by integrating it with tools like SpikeInterface (Buccino et al., 2020).
> Also, we point the reviewer and the reader at the current supplementary material figure 1 to observe a ground truth localization recovery using our method and CoM methods. Since submission, we have improved our optimization method and are now able to recover accurately and more efficiently x , y, z, and alpha on simulated data. We will include an updated version of this figure and optimization method in the revision.
>
> On the limits of the point neuron model:
>
> We acknowledge that the point neuron model is indeed simplistic yet helps us reliably and efficiently approximate the position of the action potential source. However, we can argue that aggregating the approximate localizations of spiking events for a particular neuron over the entire recording might give rise to an approximation of the overall shape of this neuron. Figure 4 shows anisotropic shapes for several units after registration, which we speculate might capture several non-somatic spiking events. We have added additional cautionary modeling discussions in the text to encourage future work on more sophisticated (and hopefully computationally efficient) localization models.
>
> We also examined the shapes of location clusters coming from spikes exhibiting model mismatch (such as asymmetry around the main channel) and it does not appear that model mismatch is a consistent source of noise in the location clusters.
>
> On DeepInterpolation:
>
> As the reviewer noted, by the time we obtain motion estimates in our pipeline, we have employed denoising at several separate stages: NN denoising the waveforms and Poisson denoising the time binned spike localizations. In contrast, these steps can be replaced by a single run of DeepInterpolation (DI). We have performed this analysis for NP2.0 data and will provide comparisons in the supplementary material. The depth/amplitude raster plot that DI produces is remarkably similar to the raster plot we have generated using Poisson denoising but at a much greater computational cost due to NN training and deployment. We have noted the DI is based on the Noise2Void method (Krull et al. CVPR 2019) and this paper has shown similar performance to BM3D as a denoising technique in several benchmarks, which our Poisson denoising internally uses after performing Anscombe transformation to Gaussianize the imaging noise. We have added this as a discussion item in the conclusion section and cited the DI paper as well as Noise2Void.

---

### Official Review · Reviewer_PJEK · 2021-07-19

**Rating:** 6
**Confidence:** 4

**Summary:**

The authors develop a new analysis pipeline for the detection and localization of spikes in Neuropixel recordings. The pipeline is based on a denoising step applied to all channels, detection of spikes via their amplitudes, triangulation across channels to localize spikes, and finally a motion correction step to counter potential movements of the electrode. The efficiency of the method is demonstrated by comparing results against several state-of-the art methods.

**Ethical Concerns:**

No concerns.

**Limitations And Societal Impact:**

Has been addressed.

**Main Review:**

The paper is well written and everything seems technically correct. The results are nicely demonstrated on various data sets, including those with (partial) ground truth data (motion). I think this paper will be of great interest to anyone recording neural activities with Neuropixel probes.

The only potential downside I see is that the proposed analysis pipeline consists largely of a sequence of analysis methods taken from the literature. While several of these methods are applied for the first time to the problem of spike localization (such as triangulation), from a purely theoretical or algorithmic point of view, the paper provides little (new) insights. In other words, the paper may not be the best fit for the Neurips audience.


**Time Spent Reviewing:**

2h

---

> ### Author Response · Authors · 2021-08-10
> **Response to reviewer PJEK**
>
> We thank the reviewer for praising the technical correctness of the paper as well as seeing its utility for the neuroscience community - especially the rapidly growing community of Neuropixels practitioners. However, as we state in the general response, we respectfully disagree with the assessment that this paper is not a good fit for the neurips community. Neurips has a long and impressive history of publishing strong neuroscience papers --- indeed, the Neurips 2021 Call for Papers specifically solicits applied neuroscience papers --- and specifically, many highly influential spike sorting papers appeared at Neurips (including the original Kilosort (Pachitariu et al. 2016) and YASS (Lee et al. 2017) papers, and the paper by Hurwitz et al. 2019 on spike localization that our work builds upon). Our paper fits perfectly within this tradition, and therefore we ask R1 to reconsider their score.

---

> > ### Comment · Reviewer_PJEK · 2021-08-16
> > **Response to authors**
> >
> > I recognize that these previous Neurips papers have all been about spike sorting, in a similar applied style. I therefore withdraw my reservation about 'fit for Neurips'.

---

### Author Response · Authors · 2021-08-10
**General Response to Reviewers**

We thank the reviewers for their time and helpful comments. We were happy to see that all of the reviewers agreed that the paper is “well written” and “technically correct” (R1), addresses a critical, timely, and interesting scientific question, and presents “very promising” results (R4). However, there was a bifurcation in the scores: R2/R3 rated the paper very highly (well above the acceptance threshold), but R1/R4 thought the paper belonged at a different venue than Neurips. We (unsurprisingly) disagree with the latter: Neurips has a long and impressive history of publishing strong neuroscience papers --- indeed, the Neurips 2021 Call for Papers specifically solicits applied neuroscience papers --- and specifically, many highly influential spike sorting papers appeared at Neurips (including the original Kilosort (Pachitariu et al., 2016) and YASS (Lee et al., 2017) papers, and the paper by Hurwitz et al. (2019) on spike localization that our work builds upon). Our paper fits perfectly within this tradition, and therefore we ask R1 and R4 to reconsider their scores.

The main technical concerns raised by reviewers were on the efficacy and validation of neural net denoising for spike localization (R3/R4) and Poisson denoising for motion estimation and registration (R2/R4). Also, R2 wondered about the validity of the point neuron model that we employ to estimate unit positions. We have thoroughly and quantitatively addressed the questions regarding denoising on localization and motion estimation by providing a text-table of the validation results below that we will add in the revised manuscript. Furthermore, we have acknowledged the limitations of the point neuron model in the discussion section with the text we have provided below. To comment on whether the point neuron model can capture spiking events in different compartments of neurons such as axons, we point the reader to the averaged localization images in figure 4 of the current manuscript. This figure shows that our aggregated localizations over long recordings could capture elongated “unit” shapes even after registration, indicating non-somatic spiking locations. We will point this out in the revised figure caption.

Specific technical responses to individual reviewers can be found in their respective responses boxes below.

---

### Decision · Program_Chairs · 2021-09-27

**Decision:**

Accept (Poster)

**Comment:**

This work brings together a number of algorithms to provide a method to detect the 3D location of cells from data collected with new high-density electrode arrays: Neuropixels. All reviewers appreciated the analysis developed by the authors in this work. The main discussion centered around the scope of NeurIPS with respect to engineered solutions to neural imaging problems. The authors, as well as some reviewers, pointed out the historical presence of such papers at NeurIPS, as well as the stated scope. Thus I am happy to recommend this paper for acceptance at NeurIPS.